



# 100+ years of recomputed surface wave magnitude of shallow earthquakes

Domenico Di Giacomo[1,*] and Dmitry A. Storchak[1]

[1]International Seismological Centre (ISC), Pipers Lane, Thatcham, Berkshire, RG19 4NS, United Kingdom

**Correspondence:** Domenico Di Giacomo (domenico@isc.ac.uk)

**Abstract.** Among the multitude of magnitude scales developed to measure the size of an earthquake, the surface wave magnitude *MS* is the only magnitude type that can be computed since the dawn of modern observational seismology (beginning of the 20th century) for most shallow earthquakes worldwide. This is possible thanks to the work of station operators, analysts and researchers that performed measurements of surface wave amplitudes and periods on analogue instruments well before the development of recent digital seismological practice. As a result of a monumental undertaking to digitize such pre-1971 measurements from printed bulletins and integrate them in parametric data form into the database of the International Seismological Centre (ISC, www.isc.ac.uk, last access: August 2021), we are able to recompute *MS* using a large set of stations and obtain it for the first time for several hundred earthquakes. We summarize the work started at the ISC in 2010 which aims to provide the seismological and broader geoscience community with a revised *MS* dataset (i.e., catalogue as well as the underlying station data) starting from December 1904 up to the last complete year reviewed by the ISC (currently 2018). This *MS* dataset is available at the ISC Dataset Repository at https://doi.org/10.31905/0N4HOS2D.

## 1 Introduction

Since its introduction, the surface wave magnitude *MS* has been very popular and for a long period of time, before the moment magnitude *Mw* was introduced by Kanamori (1977) and Hanks and Kanamori (1979), it was considered the most reliable magnitude to estimate an earthquake size. Its popularity originated due to: 1) as opposed to the magnitude concept introduced at a local scale by Richter (1935), *MS* allows seismologists to compute magnitudes for earthquakes worldwide, including those recorded at teleseismic distances (i.e., from 20° onward), without relying on local recordings that were not available in most seismic zones; 2) thanks to the work of station operators, analysts and researchers at various observatories around the world that produced readings of surface wave data for shallow earthquakes since the beginning of the last century, *MS* can be computed (systematically) since the dawn of instrumental seismology (Fig. 1). In addition, *MS* is probably the only type of earthquake magnitude that can be computed systematically for all damaging earthquakes for the last 100+ years.

Gutenberg (1945), using measurements of amplitudes and periods of surface waves accumulated during the first 40 years of the last century, introduced *MS* as: $MS = log A + 1.656 log \Delta + 1.8$. Since then a team of researchers from Moscow and Prague further developed Gutenberg's work and proposed the formula (Kárník et al., 1962; Vaněk et al., 1962): $MS = log(\frac{A}{T})_{max} + \sigma_S(\Delta) = log(\frac{A}{T})_{max} + 1.66 log \Delta + 3.3$, where *A* and *T* are the amplitude (in $\mu m$) and period (in *seconds*) of the surface wave





train, respectively, and $\Delta$ is the distance in degrees of the seismic station from the earthquake epicentre (distance and period limits will be discussed in the next section). This is the so-called Moscow-Prague formula and it was accepted as the standard for *MS* computation by the International Association of Seismology and Physics of the Earth's Interior (IASPEI, http://www.iaspei.org/, last access: August 2021) at the 1967 Zürich meeting (Bormann et al., 2012; IASPEI, 2013). The calibration

function $\sigma_S(\Delta)$ and its best fit up to 160° ($1.66 log\Delta + 3.3$) are shown in Fig. 2.

Several earthquake catalogues that listed *MS* have served the seismological community for various purposes in the past decades. One that has been instrumental for many studies is Abe's catalogue (Abe, 1981; Abe and Noguchi, 1983a, b; Abe, 1984). This catalogue lists *MS* values for large earthquakes (mostly *MS* > 6.5) up to 1980 and its reliability was recently confirmed by Di Giacomo et al. (2015a). Since then researchers have extend Abe's catalogue beyond 1980 with *MS* solutions

from the International Seismological Centre (ISC, www.isc.ac.uk, last access: August 2021) and/or the National Earthquake Information Center of the USGS (https://earthquake.usgs.gov/earthquakes/search/, last access: August 2021). Such a composite *MS* catalogue was then used as the magnitude basis for recent compilations such as the Centennial Catalogue (Engdahl and Villaseñor, 2002) and PAGER-CAT (Allen et al., 2009) as well as various types of research, from calibration purposes (Herak and Herak, 1993; Rezapour and Pearce, 1998) to patterns of the Earth's seismicity (e.g., Pérez and Scholz, 1984; Ogata and

Abe, 1991; Pacheco and Sykes, 1992; Pérez, 1999).

Considering the important legacy of *MS* in the seismological community, here we present a revised *MS* catalogue of over 46,000 earthquakes with $MS \geq 4.5$ and the underlying station data (files described in Section 8) used to derive *MS* for each earthquake. Hereafter we refer to the catalogue and underlying station data as ISC *MS* dataset (International Seismological Centre, 2021d). To create this product we benefit from the work done by Di Giacomo et al. (2015b, 2018) to digitize a large

volume of surface wave parametric data prior to 1971 and by Storchak et al. (2017, 2020) to rebuild the ISC Bulletin from 1964 onwards.

We first recall the basic steps in our procedure to compute *MS* and outline the major features of the station data behind the calculation of the network *MS*. Then we discuss some properties of the ISC *MS* dataset in terms of completeness and rates in different time periods. Finally, we briefly discuss the largest earthquakes ever recorded and outline further activities that could

improve this dataset in different time periods.

## 2 Recomputing *MS*

Our approach to computing *MS* closely follows the standard ISC procedure (Bondár and Storchak, 2011) and is already detailed in Di Giacomo et al. (2015a). However, it is beneficial here to 1) recall some aspects of the procedure in light of the content of station data files (Section 8), and 2) explain some necessary deviations from it.

First, we consider the surface wave data belonging to a reading (in ISC jargon a reading groups all parametric data from a single station associated to a specific seismic event and reported by the same agency). A reading can have any number of surface wave data entries and different reporters may provide a reading for the same station. An example of a reading is shown in Table 1 for station CLL (Collm, Germany) for an earthquake which occurred in the Northern Mid-Atlantic Ridge, 24 September





1969. We have chosen this example as the reading lists multiple surface wave data entries on all three components. Within

the surface wave phases of the reading ($L$ in our example), we first search for the maximum of $\frac{A}{T}$ on the vertical component, and, if available, the component magnitude $MS_Z$ is obtained via the Moscow-Prague formula. Then, for periods within $\pm 10$ seconds of $T$ on the vertical component, the maximum of $\frac{A}{T}$ for the horizontal vector component $\sqrt{(\frac{A}{T})_N^2 + (\frac{A}{T})_E^2}$ is searched to calculate the component $MS_H$ magnitude. If one of the two horizontal components is not available then $(\frac{A}{T})_H = \sqrt{(2*\frac{A}{T})_{N|E}^2}$. In our CLL reading example the maximum $\frac{A}{T}$ on the vertical component is defined by *ampid* = 601627636, whereas *ampid* =

601627639 and 601627638 on the North-South and East-West component, respectively, define the maximum horizontal vector component. Such defining entries are included in the station data files (more details in International Seismological Centre, 2021d). Then the *MS* for the reading is computed as $(M_{SZ} + M_{SH})/2$ if both exists, or $M_S = M_{SZ|H}$ if one of them is not available. If more than one reading *MS* exists for a station, the median of the readings *MS* is used as station *MS*. Finally, the network *MS* is computed as the median of the stations *MS* if at least three or five station magnitudes are available prior or

since 1971, respectively. The uncertainty of the network *MS* is expressed as standard median absolute deviation (SMAD) of the $\alpha$-trimmed station magnitudes ($\alpha$ = 20%).

In line with IASPEI recommendations (IASPEI, 2013), we only allow *MS* for earthquakes with depth $\leq 60$ km. The locations adopted in this work come from the ISC-GEM Catalogue (Bondár et al., 2015; Di Giacomo et al., 2018) between 1904 and 1963 and the rebuilt ISC Bulletin (Storchak et al., 2017, 2020) from 1964 onward.

Standard procedures at the ISC consider surface wave periods between 10 and 60 seconds and distances between 20° and 160°. Such delta-period ranges are also adopted here for earthquakes which occurred after 1963 (hereafter also referred to as standard delta-period ranges). Prior to 1964 we expand the period and distance ranges to 5-60 seconds and 2°-180°, respectively, as discussed in Di Giacomo et al. (2015b). The augmentation of the delta-period limits prior to 1964 is mainly due to the relative scarcity of surface wave data in the first part of the last century compared to its second half (hence the need for not discarding

station *MS*), and to changes in seismological practice in many institutes coinciding with the introduction of the World-Wide Standardized Seismograph Network (WWSSN, Oliver and Murphy, 1971; Peterson and Hutt, 2014). When stations beyond 160° are used we use the tabulated values of $\sigma_S(\Delta)$ instead of its best-fit (Fig. 2), as recommended by Bormann et al. (2012). In the next section we show that the amplitude/period measurements prior to the WWSSN introduction justifies our delta-period expansion for pre-1964 earthquakes.

## 3   Station data

A big part of the ISC mission consists of collecting and reprocessing reports from seismological agencies all over the world to produce the ISC Bulletin (International Seismological Centre, 2021c). Details about agencies contributing data to the ISC can be found at http://www.isc.ac.uk/iscbulletin//agencies/, last access: August 2021. The summary of the agencies (hereafter also referred to as reporters or data contributors) that contributed surface wave parametric data to create the ISC *MS* dataset is

shown in Fig. 3. A few aspects are worth mentioning regarding the surface wave data reporters.



Originally, the ISC had no surface wave data available in digital form for pre-1971 earthquakes. Hence, to fill this data gap, an onerous undertaking of digitizing surface wave data from station/network printed bulletins began in 2010 (Di Giacomo et al., 2015b, 2018). As shown in Fig. 3, this effort resulted in the ISC having digitized surface wave data from a total of 282 stations for over 12,000 earthquakes (it is our intention to continue this effort, see Section 6).

Between 1971 and 1998 the ISC Bulletin contains surface wave data from 457 stations worldwide. However, in this time period we cannot associate such data to specific reporters (hence reporter = UNK, unknown, in Fig. 3). The only exceptions to that are data reports (e.g., agency MOS, JEN, CLL) parsed in the ISC Bulletin during the rebuild project (Storchak et al., 2017, 2020). Since 1999, coinciding with a major update in ISC data collection procedures and the setup of the ISC database, we are able to routinely associate station data with their agency. Only 30 reporters out of about 150 contributed with surface

wave data in the last 20 years, with the largest contributors being IDC, NEIC, MOS and BJI (Fig. 3).

   The decadal spatial distribution of the stations contributing to the ISC *MS* dataset is summarized in Figs. 4-5. At times we mention seismic stations that, for sake of brevity, we may only identify by their code (station's full details can be accessed at International Seismological Centre, 2021b).

   Not surprisingly, the *MS* network geometry is unbalanced as the Northern hemisphere features many more stations than the

Southern one (a known issue in every aspect of instrumental seismology). In more detail, these figures highlight how the *MS* network became more dense and widespread over time after most of the stations were located in Europe at the beginning of the last century. Indeed, most of the *MS* in the first two decades of the last century heavily rely on stations in Germany (e.g., GTT, JEN), UPP in Sweden, and a few others (e.g., DBN in Netherlands and PUL in Russia). From the 1920s to the 1960s the station density increased in Europe and in former Soviet Union territory. North American stations also contributed but for a

small number of earthquakes.

   The Southern hemisphere had only a handful of *MS* reporting stations up to the 1970s-1980s. However, thanks to the extraordinary efficiency in observatory practice at the Observatorio San Calixto (LPZ, Bolivia, opened in 1913, Coenraads, 1993) and Riverview (RIV, Australia, opened in 1909, Drake, 1993), both from the Jesuit network (Udias and Stauder, 1996), our capabilities of obtaining *MS* improved significantly in the first half of last century both for Southern hemisphere and

worldwide earthquakes, as was noted by Gutenberg and Richter (1954).

   From the 1970s, when surface wave data started to be digitally available in the ISC Bulletin, we witness a significant increase in the *MS* network coverage, particularly in the last two decades, where many more stations in the Southern hemisphere have contributed to *MS*. However, their spatial distribution is not yet as dense as in North America or the Euro-Mediterranean area.

   To summarize the evolution of the *MS* network over the decades, Fig. 6 shows the network *MS* decadal box-and-whisker

plot of the number of stations (*Nsta*) and secondary gap (i.e., the largest azimuthal gap filled by a single station). The latter parameter is normally used as a network geometry parameter in earthquake location (Bondár et al., 2004), but here it is used as a measure of the azimuthal coverage of the station contributing to *MS* computation (both gap and secondary gap are included in the *MS* catalogue file). Ideally, the station distribution should sample the focal sphere from different azimuth to reduce the effects of the radiation pattern (von Seggern, 1970) on the network *MS*. In light of the station distributions shown in Figs. 4-5,

it is not surprising that for most of the last century the secondary gap is usually 180°-270° or above, meaning that the stations





contributing to the network *MS* are often located in a narrow azimuth. However, significant improvements occur from the 1970s, and with the increase in *Nsta* we observe an overall decrease in secondary gap.

The final aspect of the station data we discuss here regards the period at which the amplitudes of the surface waves are measured. We do that by showing, similarly to Bormann et al. (2009, 2012), the distance-period distributions of $(\frac{A}{T})_{max}$ for

earthquakes prior to and since 1964 (Fig. 7 and Fig. 8, respectively). The separation in these two time periods is linked both to the start of the original ISC Bulletin (Adams et al., 1982) in 1964 and a change in observatory practice by many institutions due to the WWSSN introduction in the early 1960s. The standard WWSSN practice produces amplitudes of surface waves as measured for *T* around 20 seconds (usually $\pm 2$ or $\pm 3$ seconds) for distances $\geq 20°$ (in addition, measurements on the vertical component were preferred to horizontal ones since the 1970s). Before WWSSN, however, the standard practice was to measure

the surface wave amplitudes in broader period ranges (such differences led IASPEI, 2013, to recommend the computation of two types of *MS*, $MS_{20}$ and $MS_{BB}$). Therefore, before 1964 we observe in Fig. 7 that *T* falls reasonably well within the expected period ranges of Vaněk et al. (1962) (i.e., amplitudes measured over a broad *T* range and using data below 20°), whereas from 1964 onward we see surface amplitudes predominantly measured around *T* of 20 seconds throughout the entire distance range, as shown by the vertical component of Fig. 8. The surface wave amplitude-period measurements pre-WWSSN, therefore, allow

us to expand the delta-period limits for pre-1964 earthquakes as outlined in Section 2.

However, not all reporters fully adopted WWSSN standards. Indeed, among the largest ones (Fig. 3), agency BJI, MOS and PRU report surface wave amplitudes in broad period ranges. The delta-period plots of those agencies are shown in Appendix A (Figs. A1, A2, A3). Other agencies, instead, strictly adhere to amplitude-period measurements around 20 seconds (Figs. A4, A5, A6, for agencies IDC, LDG and NEIC from 2009, respectively).

As a final remark in this section, we reiterate, as already done in Di Giacomo et al. (2015a), that the differences in distance and period ranges do not introduce a discontinuity in the *MS* estimates before-after 1964. The expansion of the delta-period limits pre-1964 is allowed by the data and it often gives us the opportunity to increase *Nsta* for our network *MS* computation in a time period where surface wave data was scant (compared to current times) and not digitally available (hence the need of not discarding precious and hard to get data). As a result of our approach, about 40% of the pre-1964 earthquakes we list

in the ISC *MS* dataset gained from 1 to 28 station magnitudes, and 1,000 of those earthquakes would not have network *MS* without delta-period augmentation. This is synthesized in Fig. 9. An area encompassing the North Atlantic mid-oceanic ridges, the Euro-Mediterranean and the Middle-East benefitted the most thanks to European and central Asian stations that measured surface waves in broad period ranges at distances below 20°.

## 4 Catalogue properties

The ISC *MS* dataset has a minimum cut-off magnitude of 4.5. Earthquakes with lower *MS* values are available in the ISC Bulletin but mostly in recent decades. The major improvements regard earthquakes prior to 1964, where, according to our records, out of 10,057 earthquakes the ISC is the first to compute *MS* for 4,940 of them (their distribution and timeline is shown in Fig. A7).





Considering the whole ISC *MS* dataset, major features can be discussed using Fig. 10, where we show the magnitude time-
line, number of earthquakes per year for various magnitude thresholds and annual magnitude of completeness (*Mc*) computed
with the maximum curvature method of Wiemer and Wyss (2000). Overall, we include *MS* < 5.5 earthquakes mostly from the
1980s, and *Mc* approaches approximately 4.5 in the last 20 years. We note that we were able to obtain more solutions at the
low magnitude end particularly in the late-1920s-1930s. This has been possible thanks to the establishment of the backbone
network in former Soviet Union territory and a general increase of *MS* stations in other areas (see annual station maps in Inter-
national Seismological Centre, 2021d). An overall dip is observed in the 1940s, most likely caused by the disruption of World
War II on the seismic network (Di Giacomo et al., 2018). Another fluctuation at low magnitudes is observed in the early-1980s.
Indeed both the annual counts and *Mc* show a significant variation from 1978-1979 (*Mc* close to 5.0) to 1980-1983 (higher
*Mc* ranging between 5.2-5.5). We believe this is due to the temporary absence of MOS surface wave data in 1980-1983 (see
Section 6), which was included into the rebuilt ISC Bulletin (Storchak et al., 2020) from 1984 onward (*Mc* dropping again to
about 5 and below).

Less strong variations are seen for moderate size earthquakes (i.e., *MS* between 5.0 and 6.0). The early part of last century
(up to the mid-1920s) is clearly complete above magnitude 6.0, whereas since the 1950s the frequency of *MS* 5.5 and 6.0
appears rather stable. Pronounced variations are observed from the mid-1920 to the 1940s for reasons mentioned above.

Variations over time of the frequency of large (i.e., $MS \geq 6.0$ ) earthquakes based on past catalogues have been the subject of
debate in past literature. In particular, Pérez and Scholz (1984) suggested that, under the assumption of constant rate earthquake
occurrence, temporal variations of large shallow earthquakes were driven by instrumental changes. Ogata and Abe (1991) and
more recently Ogata (2021), however, suggest that variations in the frequency of global large earthquakes are a real effect of
the Earth's seismic activity (long-range dependence nature of earthquake occurrence). Therefore, to further discuss the rate
of the Earth's large shallow seismicity, we show in Fig. 11 the cumulative number of strong to major earthquakes in the ISC
*MS* dataset similarly to the figures in Pérez and Scholz (1984) and Pérez (1999). Compared to these works, our rates for *MS*
$\geq 7.0$ and 6.0 in different time intervals (Table 2) show some significant differences and lack large jumps from one period to
another. This is strikingly evident for the $MS \geq 6.0$ distribution, where Pérez (1999) rate goes down to $38y^{-1}$ during 1964-1978
compared to a rate of about $78y^{-1}$ in the ISC *MS* dataset. We note that this period in the original ISC Bulletin lacked the ISC's
own computations of *MS*. Therefore, Pérez (1999) rates may have been biased by using largely incomplete inputs.

In general, we see that shallow seismicity rates are characterized by a global low occurring between the great earthquakes
of the early 1960s and the beginning of the current century (Ammon et al., 2010). Although rates for $MS \geq 7.0$ in the first part
of the last century are comparable to the rate we have observed since 2005, it seems that rates for $MS \geq 6.0$ from the WWSSN
introduction appear to be lower than rates in the first part of the last century. We also assessed if by declustering (Reasenberg,
1985) the *MS* catalogue the rates would be different but only small variations occur, and, more importantly, relative differences
between time periods remain. This is not surprising as the ISC *MS* dataset does not contain a large number of aftershocks for
$MS \geq 6.0$ (by the very nature of *MS* it is more difficult to obtain it for aftershocks of large earthquakes due to association
challenges in overlapping signals, particularly in routine operations).





It is not the aim of this work to investigate whether fluctuations in seismic activity rates are partially due to instrumental changes or purely due to natural variations of the Earth's seismicity. However, we believe that the ISC *MS* dataset is one of the best inputs to date to do such studies. In this context it is important to point out that the quality of instrumental earthquake catalogues depends on the quality of the data available at the time of processing. In our experience, for long-term datasets it is almost unavoidable that different types of shortcomings may occur in different time periods, for example due to external factors (e.g., network deficiencies during World Wars) and that faulty individual entries may be present. It is of paramount importance, therefore, that datasets are well-documented and that users know how they are created in order to properly use them for research.

## 5 On the *MS* saturation and large differences with *Mw*

By the time *MS* was introduced by Gutenberg (1945) no magnitude 9 or 9+ earthquake had been recorded instrumentally. The occurrence of the 4 November 1952, Kamchatka earthquake and the well-know great earthquakes of the early 1960s (22 May 1960, Chile and 28 March 1964, Alaska earthquakes), drew attention to a shortcoming of *MS*, that is commonly referred to as magnitude saturation. This was one of the factors that led Kanamori (1977) and Hanks and Kanamori (1979) to introduce the moment magnitude *Mw*, which is based on a physical parameter of the seismic source (i.e., seismic moment) rather than amplitude-period measurements.

In Fig. 12 we compare the ISC *MS* dataset with *Mw* from GCMT and the bibliographic search for pre-1976 earthquakes of Lee and Engdahl (2015) and follow-up updates as listed at www.isc.ac.uk/iscgem/mw_bibliography.php, last access: August 2021 (hereafter referred to as *Mw* from literature). Such magnitude comparison has been discussed in several papers, particularly to derive magnitude conversion relationships. For this work, however, we show this comparison to focus on the *MS* saturation issue and briefly touch upon earthquakes with large (*Mw-MS*) differences.

The ten largest earthquakes (in *Mw* terms) ever recorded are easily identified in Fig. 12 by the event code in the ISC Event Bibliography (Di Giacomo et al., 2014; International Seismological Centre, 2021a). As already summarized by Kanamori (1983), the saturation of *MS* is generally expected to start between 8.2 and 8.5. For recent (26 December 2004, Sumatra and 11 March 2011, Tohoku) and pre-GCMT earthquakes (the above mentioned Chile 1960 and Alaska 1964) with *Mw* 9 and above, the effects of saturation are quite severe and vary between 0.6 and 1 magnitude unit (m.u.). However, for earthquakes with *Mw* between 8 and 9 the variation of the saturation appears to vary much more (from near to 0 up to 1 m.u.). For example, the 27 February 2018, Maule, *Mw* 8.8 earthquake has an *MS* of only 0.25 m.u. smaller, close to common *Mw-MS* differences observed across a wide magnitude range before the saturation of *MS* is expected. Fig. 12 shows other examples where *Mw* and *MS* are close to the 1:1 line between 8 and 8.7 (including the 15 August 1950 Assam earthquake). On the other hand, large differences are observed for other earthquakes (e.g., 4 February 1965, Rat Islands and the 28 March 2005, Nias earthquakes). This often occurs to a peculiar category of events, the so-called tsunami earthquakes (Kanamori, 1972). As already well-documented in the literature, these are earthquakes characterized by a relatively small *MS* compared to their *Mw*. The most striking example





is probably the 1 April 1946, Aleutian earthquake, where our *MS* of 7.4 is much smaller than the *Mw* 8.6 by López and Okal (2006).

Considering the *MS* values for the largest earthquakes ever recorded, we support the remarks by Bormann (2011) that it would be more correct to speak of *MS* underestimation rather than saturation, as the latter would require to be systematically observable for great earthquakes with *Mw* between 8.2-8.5 and above. However, we have shown that underestimation ("satu-

ration") depends on the type of earthquake, and it is severe only for a handful of the largest earthquakes ever recorded (*Mw* ≥ 9). Therefore, the underestimation ("saturation") of *MS* should not discourage researches to use it as a reliable measure of the size of shallow earthquakes.

Overall, the magnitude comparison of Fig. 12 shows that *MS* is typically close to *Mw* over the magnitude range 6.2 to 8, whereas for smaller earthquakes *MS* is usually smaller than *Mw*. However, some earthquakes show large differences (examples

listed in Table A1, for *MS* ≫ *Mw* and vice-versa). For the sake of brevity we do not discuss every earthquake with such large differences but touch only on the case of the 18 April 1906, San Francisco earthquake (SANFRANCISCO1906, first event in Table A1) to clarify our reasons for keeping such entries in the ISC *MS* dataset.

The *MS* of the SANFRANCISCO1906 earthquake is dominated by stations located in Germany (GTT, POT, LEI and JEN), plus Apia (API, Samoa Islands) and OSA (Osaka, Japan) at different azimuths. All station *MS* are consistently above 8,

resulting in a network *MS* of 8.6±0.1 (full station details listed under event = 16957905 in International Seismological Centre, 2021d). This is a much higher value than the *Mw* = 7.7 obtained by Wald et al. (1993). Such outliers occur in most earthquake catalogues for various reasons. As mentioned earlier, parameters of individual earthquakes are the result of the processing of the data available at a given time. For the SANFRANCISCO1906 earthquake, instrumental issues could have played a major role in the high *MS* value. However, we believe that listing such results in the dataset (rather than deprecating) is important

for legacy reasons, and that users may still use such information for further studies and, ideally, motivate the community to attempt additional data collection. The latter is an activity that we will continue and discuss in the next section.

## 6 Future developments

The maintenance and development of the ISC *MS* dataset will not cease with this work. First, we intend to routinely add the last calendar year reviewed by the ISC. This means that once the ISC review is over for 2019 earthquakes, the ISC *MS* dataset will

be updated and end in 2019, and so on in following years. Secondly, we aim at refining and adding *MS* solutions for past years. Indeed, we are aware that the *MS* station contribution can be improved in certain years. One example was already pointed out for 1980-1983, where MOS surface wave data was not included in time for ISC rebuild project (Storchak et al., 2020). By adding such data we expect to fill (or, at least, partially fill) the gap shown in those years for low *MS* earthquakes, as shown in Fig. 10.

Before the early 1980s, the following time periods may benefit from additional station contributions:





- During the 1970s, one source that, to our knowledge, has never been digitized, is the printed bulletins of the Chinese network. Tens of stations with plenty of surface wave data are available in those bulletins, which can potentially increase *Nsta* for earthquakes already listed in the dataset and allow us to compute *MS* for several new ones;

- Due to time and funding limitations, the digitazion of 1964-1970 surface wave data from printed station/network bulletins (Di Giacomo et al., 2015b) was not done for all bulletins available at the ISC, and, if done, it focused on earthquakes with magnitude 5.5 and above. Hence, a more comprehensive approach for surface wave data digitazion is desirable for these years;

- For the period 1936-1963 we are finalizing the digitazion of station arrival times for earthquakes in the bulletins of the Bureau Central International de Séismologie (BCIS, 1933-1968) that were not listed in the International Seismological Summary (ISS, 1918-1963) (earthquakes in this time period that are not listed in the ISS currently have no station data digitally available in the ISC database). Once this undertaking is finished, we will add surface wave data for earthquakes recorded teleseismically and attempt to obtain *MS* for as many earthquakes as possible;

- Improvements in the first part of the last century are more challenging as we have nearly exhausted the digitazion of printed bulletins available to us. It is hard to verify if our surface wave data collection from printed bulletins is as complete as possible. Assistance in this respect from observatories and archives around the world would be highly appreciated (the presence or absence of a set of station data can be easily checked in the ISC *MS* dataset).

Hence, if conditions permit, we wish to continue the digitazion of printed bulletins and add surface wave data in order to improve the *MS* solutions for a significant fraction of pre-1980 earthquakes. However, we stress that additional contributions for earthquakes in recent decades are welcome as well and we will strive to include them in the ISC Bulletin, and in turn, in the ISC *MS* dataset.

## 7 Conclusions

An aspect that differentiates *MS* from other magnitude scales is that it can be computed from original measurements of surface wave amplitudes and periods throughout the instrumental period. The ISC *MS* dataset we presented here includes 100+ years of earthquakes with $MS \geq 4.5$ starting from the dawn of modern instrumental seismology (1904) up to the last complete reviewed year by the ISC (2018). This achievement is possible as a result of a monumental undertaking thanks to which pre-1971 measurements of surface waves were digitized from a multitude of printed station/network bulletins.

We have summarized the evolution of the station network contributing to *MS* and highlighted its shortcomings (e.g., significant lack of stations in the Southern hemisphere for a large part of the last century) and strengths (e.g., high density in Europe that allowed us to obtain *MS* for earthquakes in a wide area in low magnitude ranges before the introduction of modern digital stations). The expansion of the delta-period ranges, as allowed by the data, resulted in more and better constrained *MS* estimations for about 40% of the pre-1964 earthquakes.

We have discussed the *MS* underestimation for the largest earthquakes ever recorded and pointed out the presence of occasional large differences with *Mw*. Those entries are listed for legacy and other purposes and may require further work.

Inevitably, the dataset has fluctuations in terms of completeness and earthquake rates over different time periods. We dis-
cussed the most relevant ones and outlined plans for continuing and improving this dataset.

In the years to come we envisage the ISC *MS* dataset as one of the best input researchers can use for various seismological studies, including the Earth's seismicity patterns.

## 8   Data availability

The ISC *MS* dataset (International Seismological Centre, 2021d) is available in the ISC Dataset Repository at http://doi.org/
10.31905/0N4HOS2D. It is composed of a catalogue file (CSV format) and annual files containing the underlying station data used to obtain *MS* for each earthquake. All parameters in the catalogue and annual files are detailed in the README file in International Seismological Centre (2021d). The annual files include, below the earthquake parameters, two data blocks: first the station magnitude block (sorted by distance) and then the phase data block, which includes the original amplitude and period measurements as well as the intermediate magnitude results (amplitude and reading magnitude, $MS_Z$, $MS_H$) that lead to
the station magnitude computation (see Section 2). Annual station plots and annual station lists are also included as well as the file with the data points to generate Fig. 12.

*Author contributions.*   DDG is the lead author, prepared the dataset and figures, supervised the digitazion of the surface wave data and vetted the *MS* results up to 1963. DAS obtained the funding for the work and established and maintained operational connections with many data providers, especially in obtaining additional datasets previously unavailable. Both authors contributed to the manuscript and approved the
final version.

*Competing interests.*   The authors declare that they have no conflict of interest.

*Acknowledgements.*   We are grateful to all reporters that contribute or have contributed data to the ISC, particularly in terms of *MS* for this work. Special thanks go to many colleagues that lent or donated to the ISC station/network printed bulletins originally not available to us, as summarized at www.isc.ac.uk/iscgem/acknowledge.php, last access: August 2021. Daniela Olaru and former data entry staff were
instrumental to this work for digitizing the surface wave data from printed bulletins. The ISC is able to continue its mission thanks to the support of its members (http://www.isc.ac.uk/members/, last access: August 2021) and sponsors (http://www.isc.ac.uk/sponsors/, last access: August 2021). Work partially funded by NSF grants 1811737, 1417970 and 0949072; USGS Awards G14AC00149, G15AC00202, G18AP00035 and G19AS00033. All figures were drawn using the Generic Mapping Tools (Wessel et al., 2013).





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

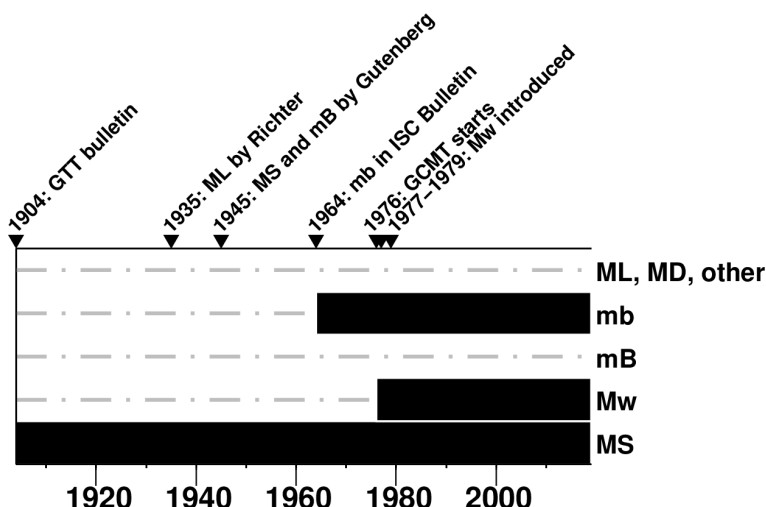

**Figure 1.** Availability over time of common magnitude scales for worldwide (*MS*, *Mw*, broad-band and short-period body-wave magnitude *mB* and *mb*, respectively) and local/regional (e.g., Richter and duration magnitude *ML* and *MD*, respectively) earthquakes. Solid thick black lines represent time periods over which a magnitude scale is available or can be recomputed systematically, dashed-dotted thin grey lines otherwise. For local/regional magnitudes the availability only regards limited continental areas (Di Giacomo and Storchak, 2016). On top are listed some significant developments in terms of earthquake magnitude. Among those GTT refers one of the first printed station bulletin produced at the Göettingen observatory in Germany (Schering, 1905), which pioneered modern observational seismological practice, and GCMT is the Global Centroid Moment Tensor project (www.globalcmt.org, last access: August 2021, Dziewonski et al., 1981; Ekström et al., 2012).

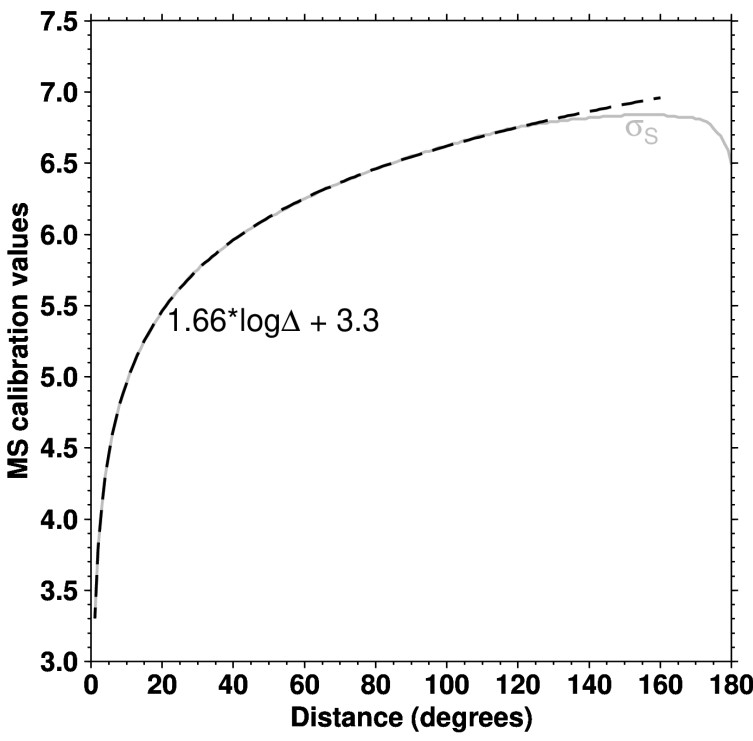

**Figure 2.** *MS* calibration function (tabulated values, $\sigma_S$) from the Moscow-Prague (grey solid curve) group and its best-fit for distances between 2° and 160° ($1.66 log\Delta + 3.3$, black dashed curve). See Bormann (2012) for details.

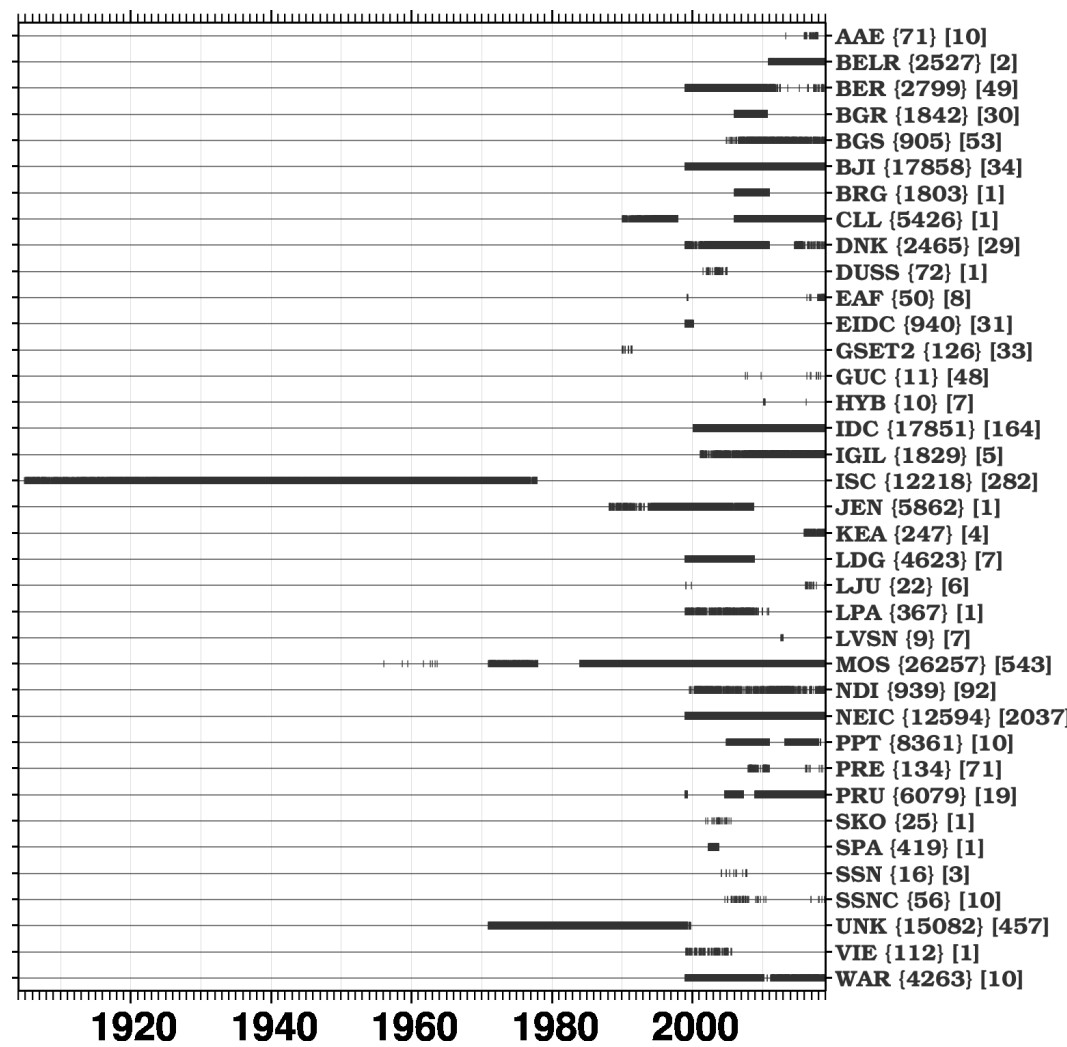

**Figure 3.** Timelines of the agencies contributing with surface wave data (amplitude and period measurements). Each symbol represents the origin time of an earthquake. Details about each agency code can be found by typing the agency code at www.isc.ac.uk/iscbulletin//agencies/, last access: August 2021. The total number of earthquakes and stations for each agency are listed in curly and square brackets, respectively. Note that reporter = UNK (unknown) is not a genuine reporter code but it simply represents data collected before the ISC database was set up, i.e., when the association between data and reporter was not maintained.

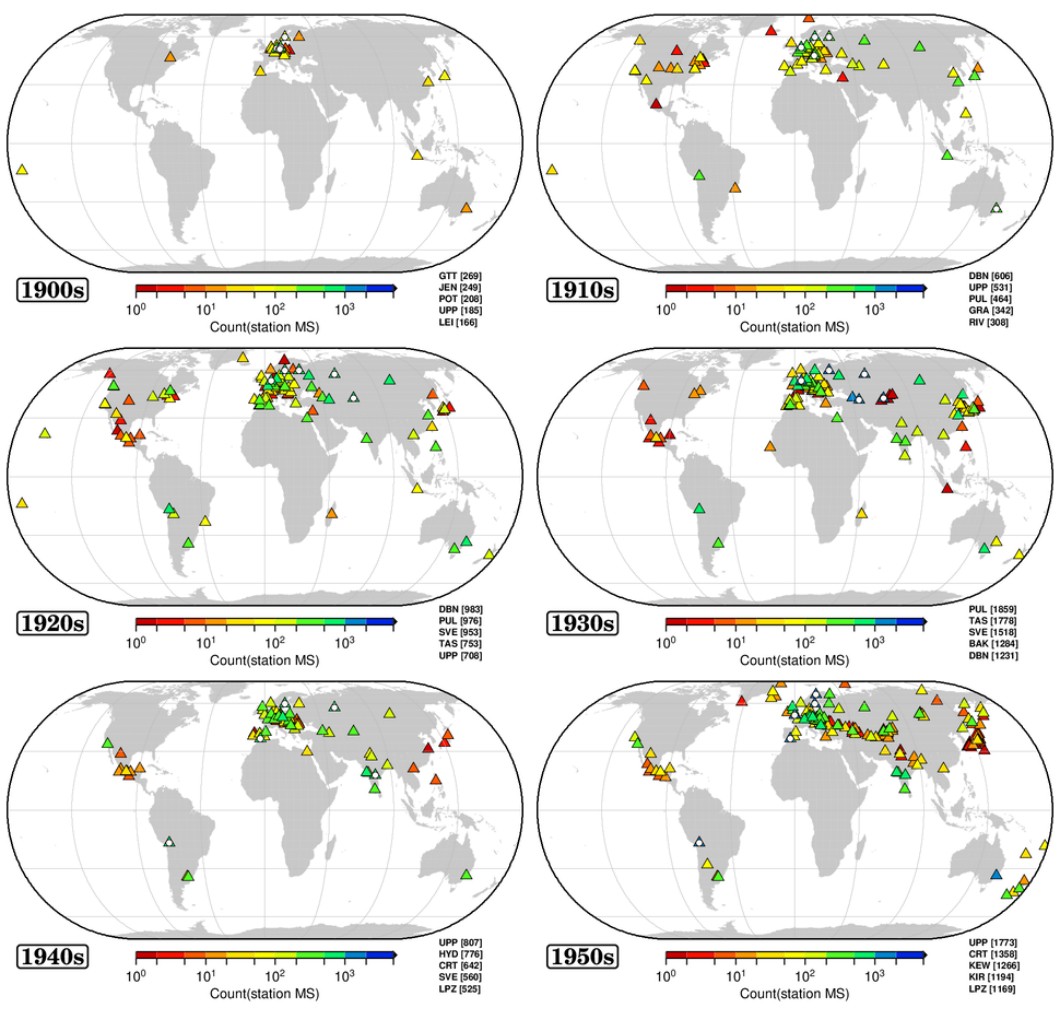

**Figure 4.** Decadal (up to the 1950s) distribution of the stations (triangles) that contributed with surface wave data. Symbols are colour-coded by number of station *MS*. For each decade, the top five stations in terms of *Count(station MS)* are identified by a white circle and listed in the bottom-right corner outside each map. Maps drawn using the Generic Mapping Tools (GMT) (Wessel et al., 2013) software. Plots of the annual station *MS* distributions are included in International Seismological Centre (2021d).

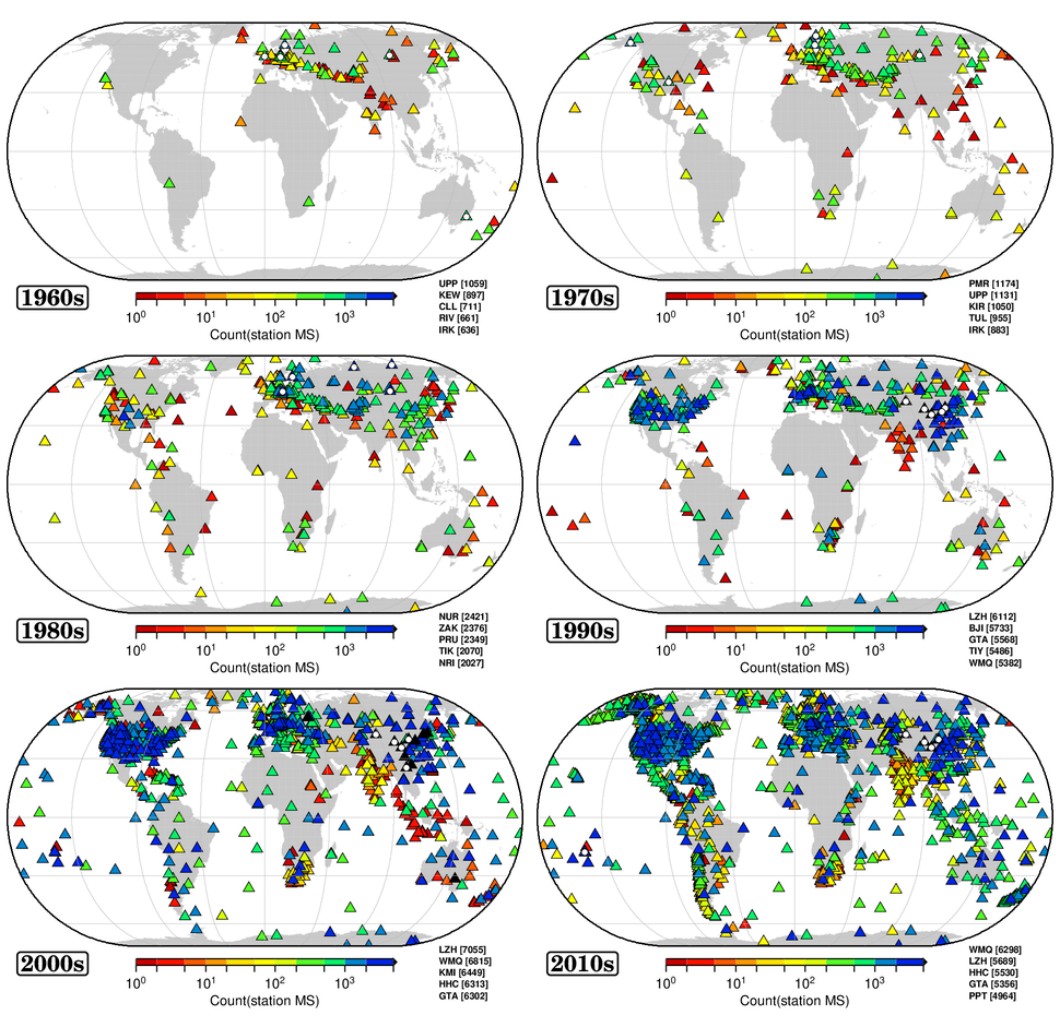

**Figure 5.** As for Fig. 4 but since the 1960s. Maps drawn using the Generic Mapping Tools (GMT) (Wessel et al., 2013) software.


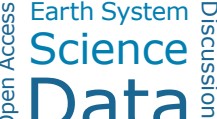

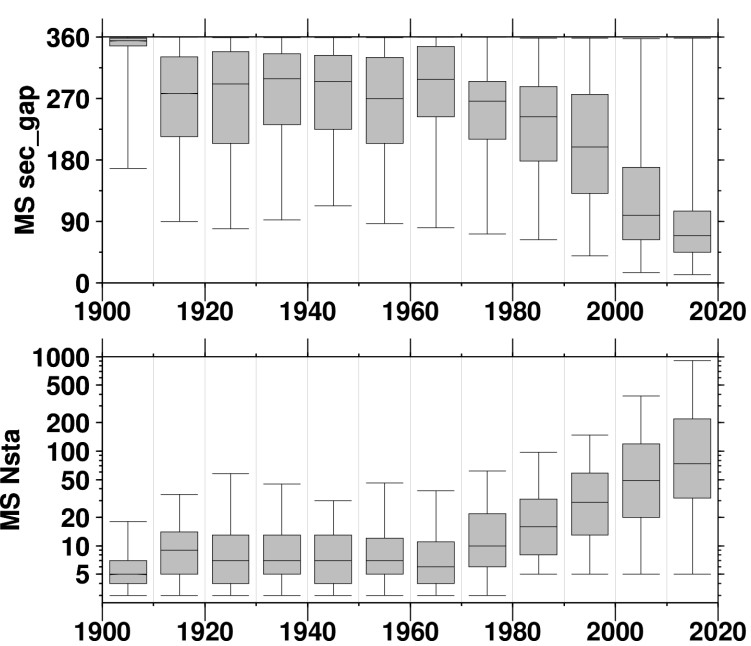

**Figure 6.** Decadal box-and-whisker plot of the secondary gap for *MS* (top) and number of *MS* stations (bottom).

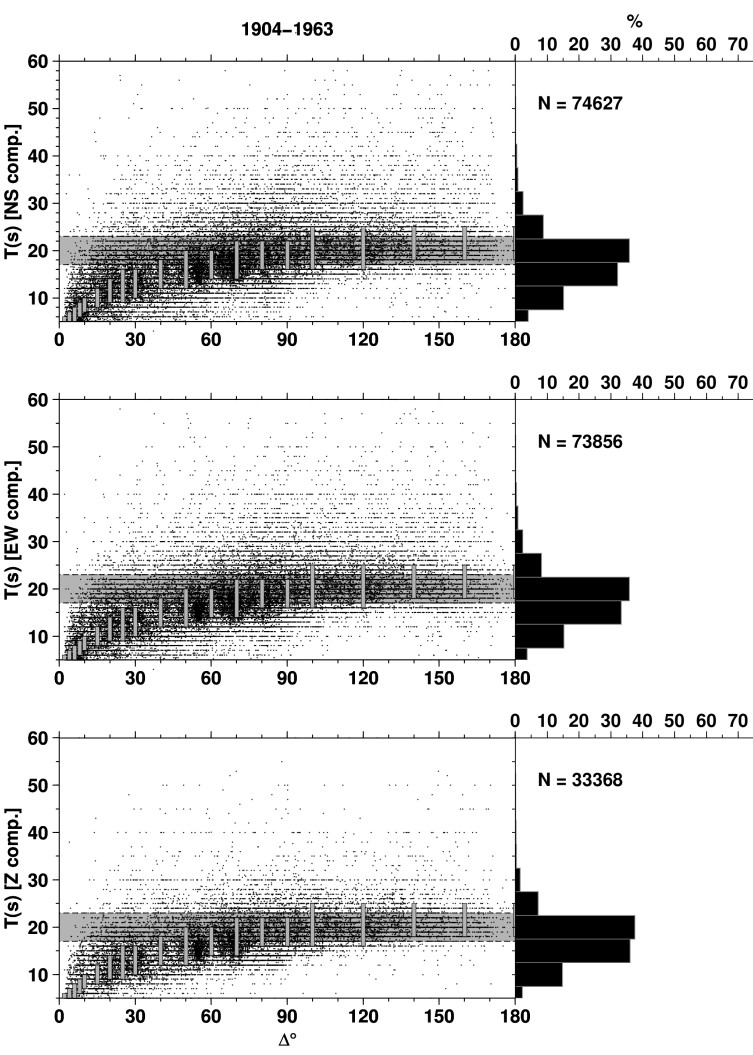

**Figure 7.** 3-component distance-period plots of $(\frac{A}{T})_{max}$ for surface wave readings digitized from printed bulletins for earthquakes that occurred before 1964. The horizontal grey shaded area depicts measurements around 20 seconds, whereas the vertical grey bars represent the expected period ranges at various distances (Vaněk et al., 1962) as published in Table 3.2.2.1 of Willmore (1979). The histograms on the right-hand side show the period distribution in bins of 5 seconds. See text for details.



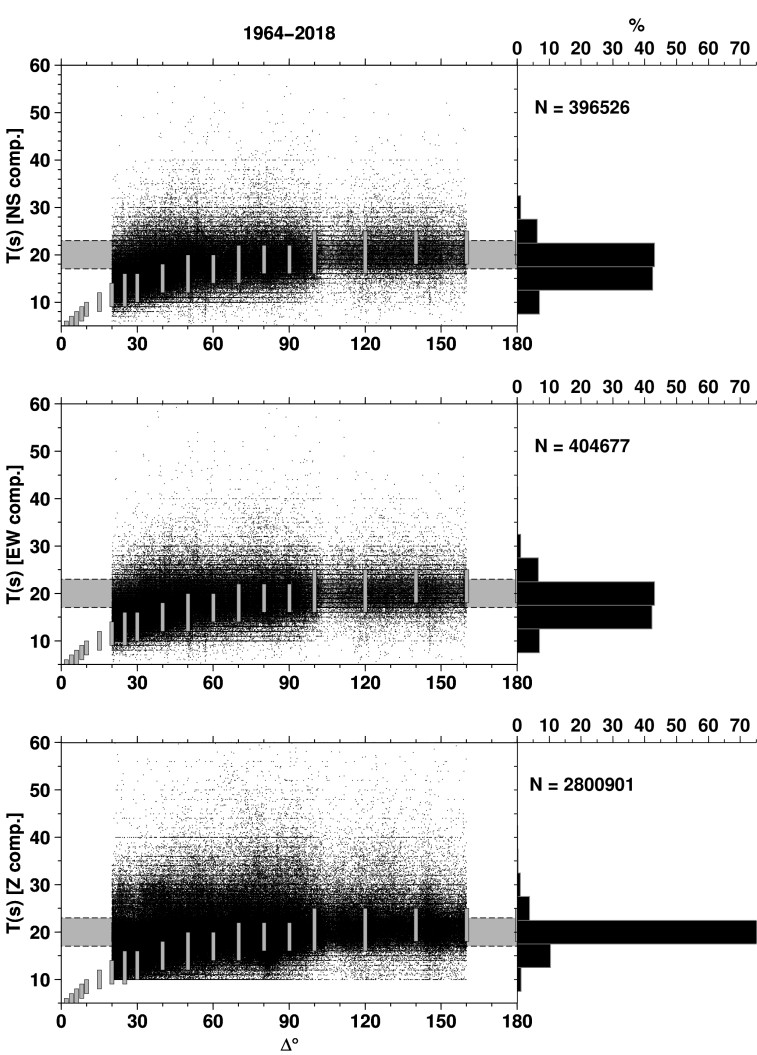

**Figure 8.** As for Fig. 7 but for 1964-2018.

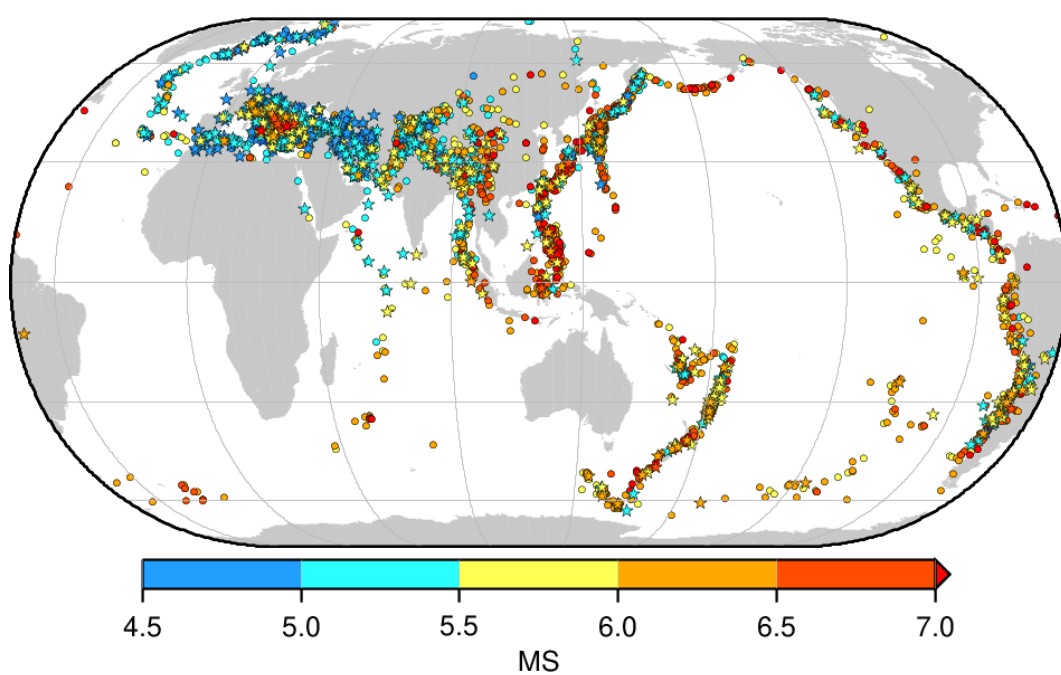

**Figure 9.** Map of the pre-1964 earthquakes where the expansion of the delta-period ranges compared to the current ISC practice allowed us to better constrain *MS*. Stars are for the 1,000 earthquakes that otherwise would not have network *MS*. All symbols colour-coded by *MS*. Map drawn using the Generic Mapping Tools (GMT) (Wessel et al., 2013) software.

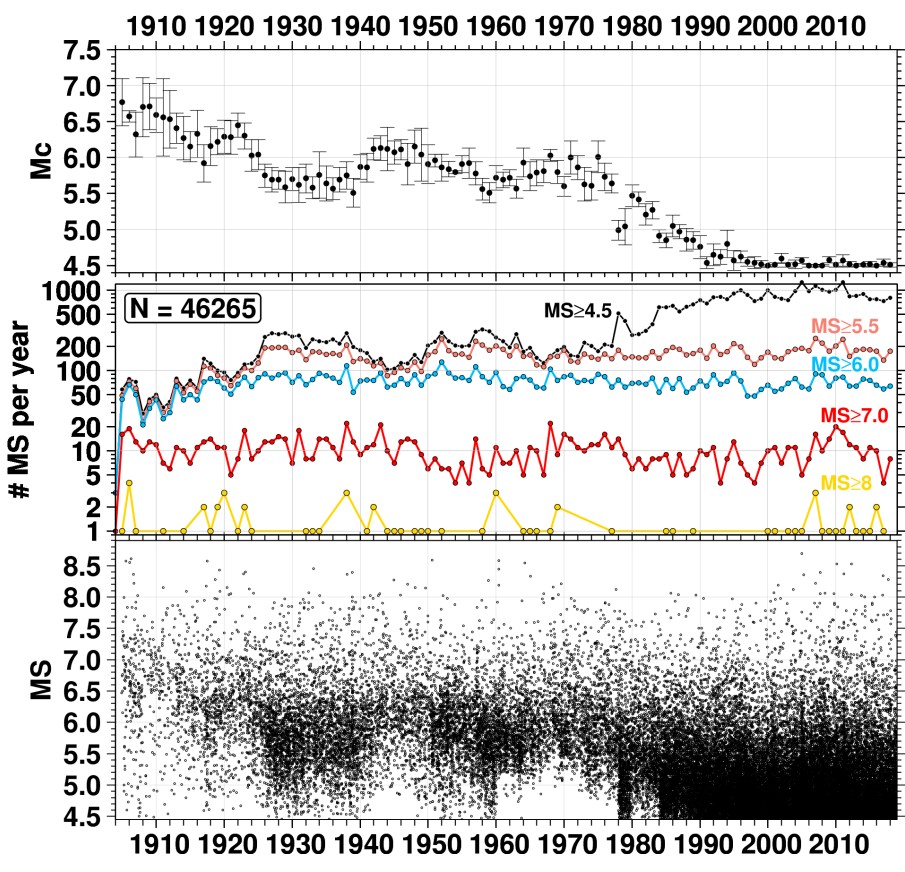

**Figure 10.** Bottom panel: magnitude timeline of the ISC *MS* dataset. Middle panel: number of *MS* per year for different *MS* thresholds (4.5, 5.5, 6.0, 7.0 and 8.0 in black, light red, blue, red and yellow, respectively). Top panel: annual magnitude of completeness *Mc* (Wiemer and Wyss, 2000) in the dataset, shown as average ± 1 standard deviation.

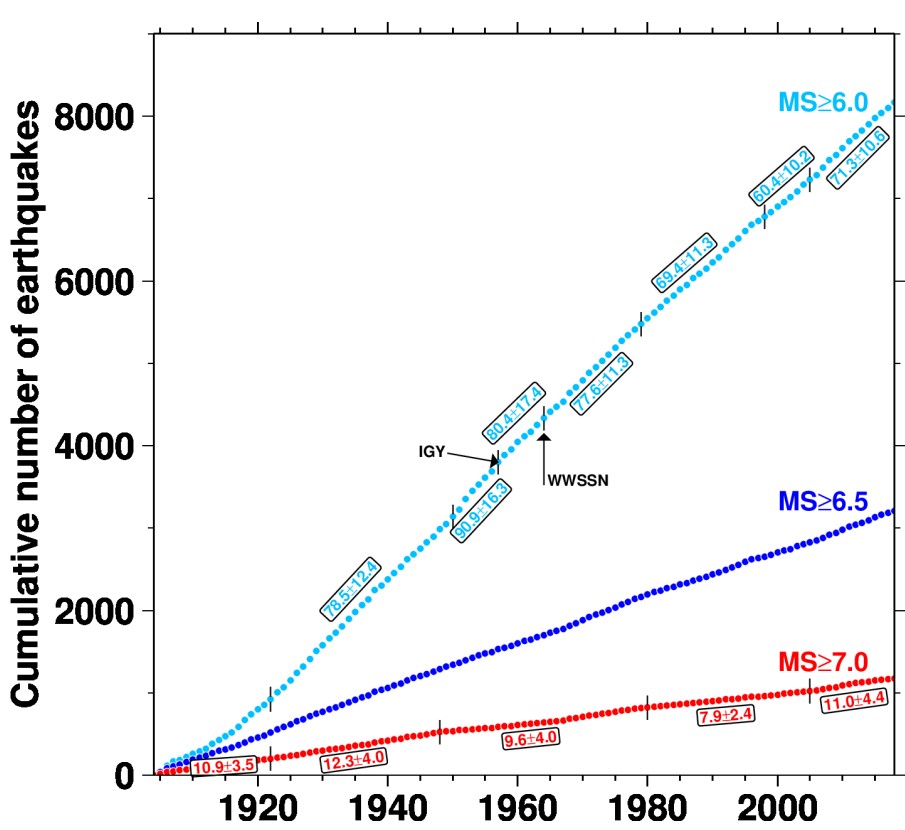

**Figure 11.** Cumulative number of earthquakes with *MS* ≥7.0 (red), ≥6.5 (blue) and ≥6.0 (cyan). The vertical black segments on the red and cyan symbols locate the time periods considered by Pérez and Scholz (1984) for *MS* ≥7.0 (up to 1980) and Pérez (1999) *MS* ≥6.0 (from 1950 to 1997), respectively. IGY stands for International Geophysical Year started in 1957. See text for details.

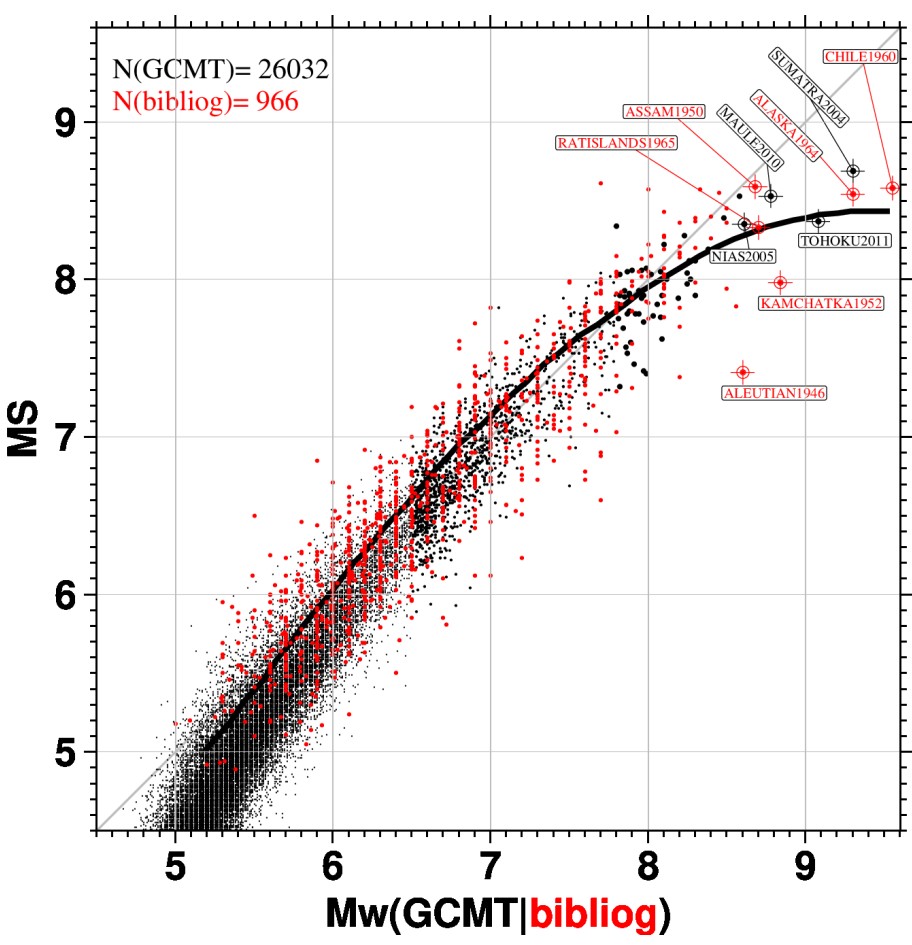

**Figure 12.** Comparison of *MS* with *Mw* from GCMT (black dots) and from the literature (red, Lee and Engdahl, 2015, and further updates listed at www.isc.ac.uk/iscgem/mw_bibliography.php, last access: August 2021). The black solid curve is a digitized version of the mid-point *Mw-MS* curve shown in Figure 4b of Kanamori (1983). The largest 10 earthquakes in terms of *Mw* (bulls eye symbols) are also identified by the event code in the ISC Event Bibliography (www.isc.ac.uk/event_bibliography/index.php, last access: August 2021).

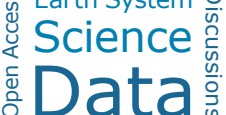

**Table 1.** Reading example from station CLL (Collm, Germany) associated to an event in the Northern Mid-Atlantic Ridge occurred the 24 of September 1969 ($\Delta = 58.8°$). The surface wave measurements (L phases) are used by our procedure to obtain the reading *MS*. *Rdid* and *ampid* are database identifiers for the reading and single amplitude-period entries, respectively. See text for details.

| Phase | Comp. | Arrival Time (UTC) | rdid | ampid | A ($nm$) | T ($s$) |
|-------|-------|--------------------|------|-------|----------|---------|
| S | N | 1969-09-24 18:21:14 | 38936 | 601627645 | 16500 | 20.0 |
| S | E | 1969-09-24 18:21:14 | 38936 | 601627644 | 25500 | 20.0 |
| L | N | 1969-09-24 18:32:00 | 38936 | 601627642 | 23000 | 24.0 |
| L | E | 1969-09-24 18:32:00 | 38936 | 601627641 | 44500 | 24.0 |
| L | Z | 1969-09-24 18:32:00 | 38936 | 601627640 | 43500 | 24.0 |
| L | N | 1969-09-24 18:39:00 | 38936 | 601627639 | 24000 | 17.0 |
| L | E | 1969-09-24 18:39:00 | 38936 | 601627638 | 30000 | 17.0 |
| L | Z | 1969-09-24 18:39:00 | 38936 | 601627637 | 25000 | 17.0 |
| L | Z | 1969-09-24 18:43:00 | 38936 | 601627636 | 32000 | 16.0 |



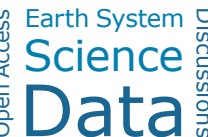

**Table 2.** Rates of seismicity in the ISC *MS* dataset and in Pérez and Scholz (1984); Pérez (1999). See text for details.

| *MS*≥7.0 | Period | Rate (ISC *MS*) | Rate (Pérez and Scholz, 1984) |
|---|---|---|---|
| | 1905-1922 | $10.9 \pm 3.5$ | $7.3 \pm 1.8$ |
| | 1922-1948 | $12.3 \pm 4.0$ | $12.6 \pm 4.0$ |
| | 1948-1980 | $9.6 \pm 4.0$ | $7.1 \pm 2.8$ |
| *MS*≥6.0 | Period | Rate (ISC *MS*) | Rate (Pérez, 1999) |
| | 1950-1956 | $90.9 \pm 16.3$ | $109 \pm 17$ |
| | 1957-1963 | $80.4 \pm 17.4$ | $137 \pm 14$ |
| | 1964-1978 | $77.6 \pm 11.3$ | $38 \pm 6$ |
| | 1979-1997 | $69.4 \pm 11.3$ | $55 \pm 7$ |





**Appendix A: Additional plots**

Here we show figures in support of the main text. Most of them regard additional delta-period plots similar to Fig. 7 for major *MS* reporters.



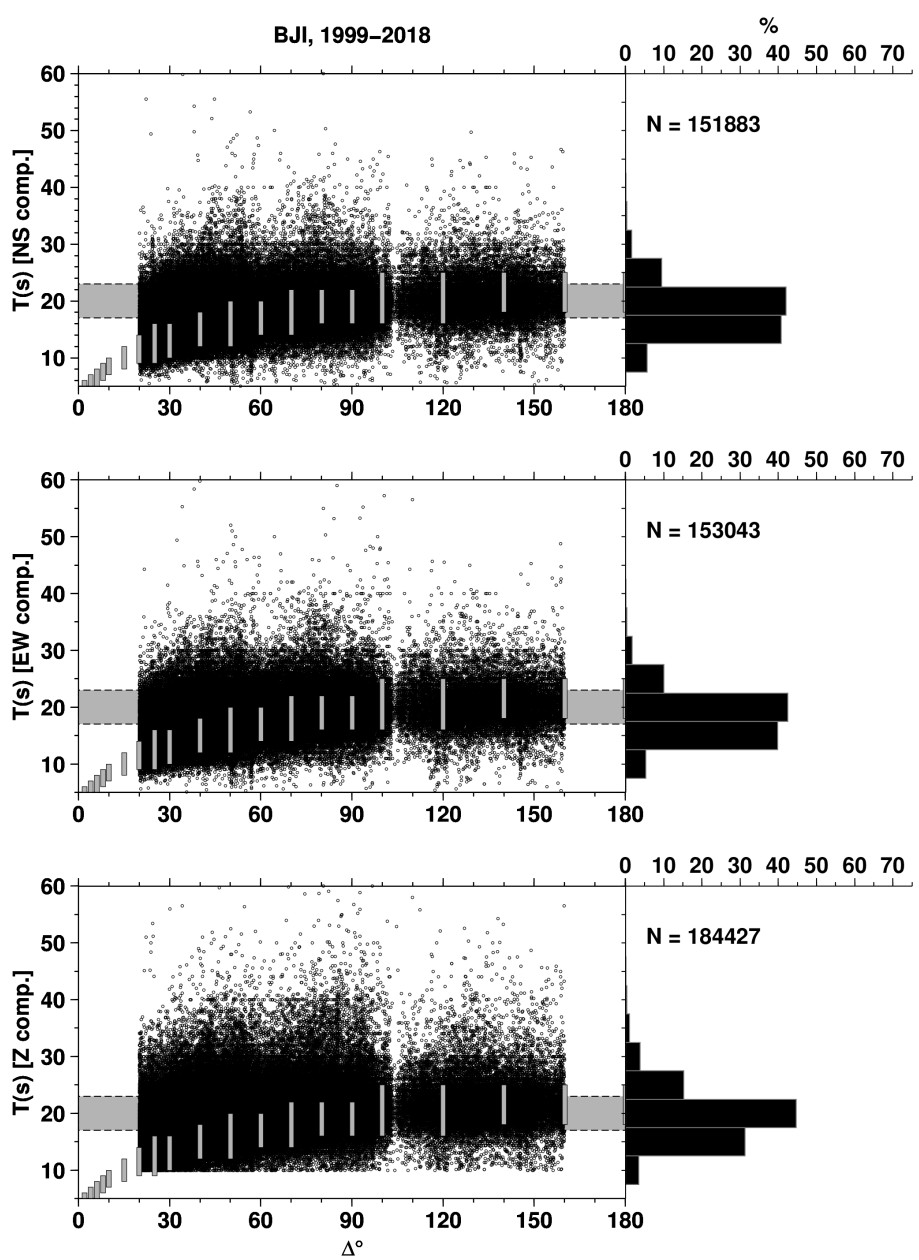

**Figure A1.** As for Fig. 7 but for reporter BJI.

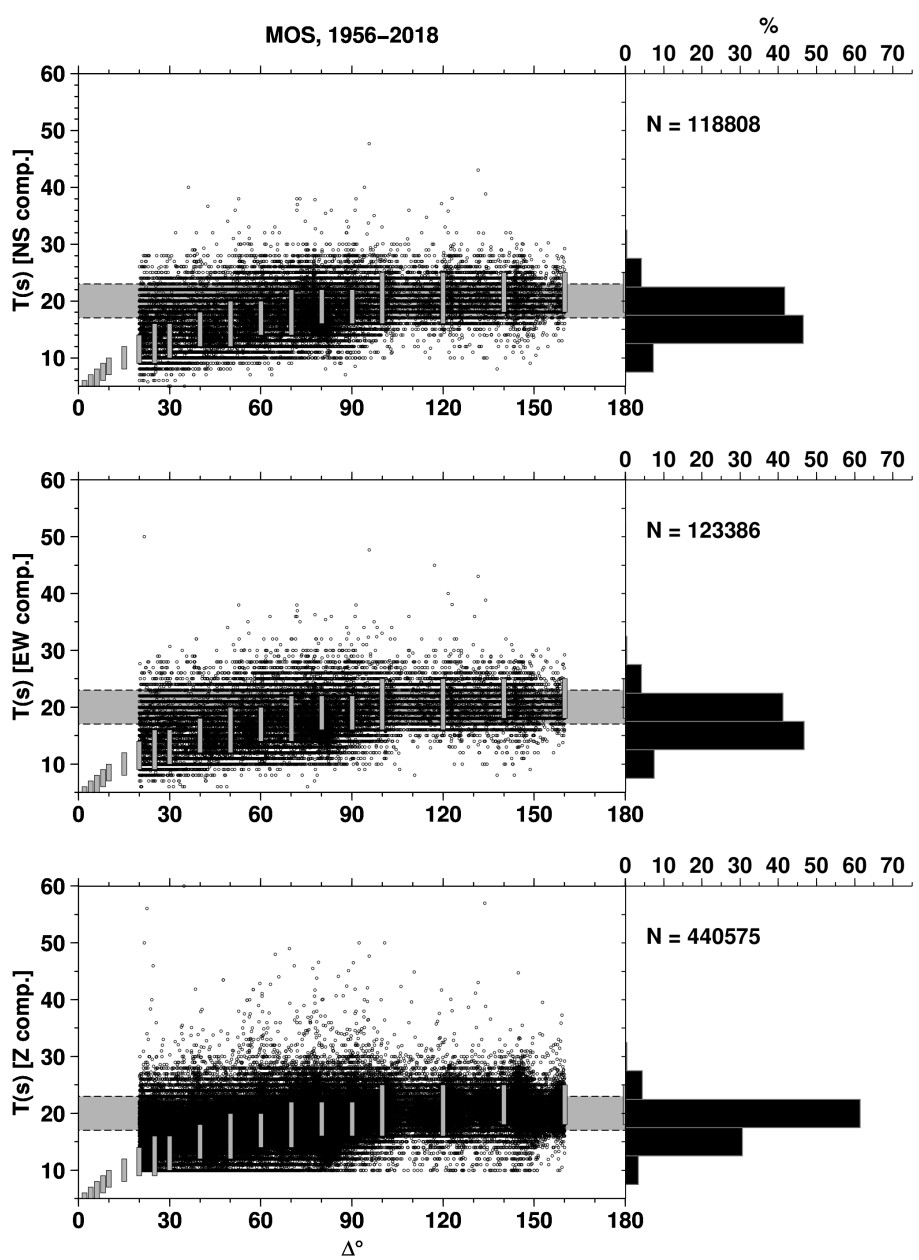

**Figure A2.** As for Fig. 7 but for reporter MOS.



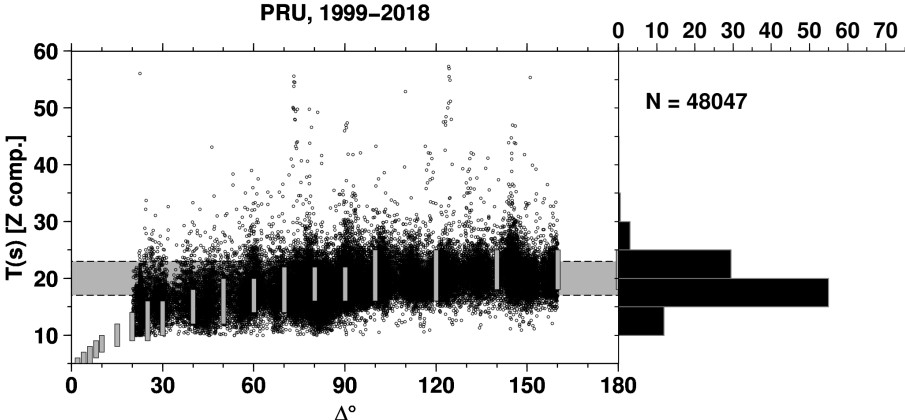

**Figure A3.** As for Fig. 7 but for reporter PRU, vertical component only.



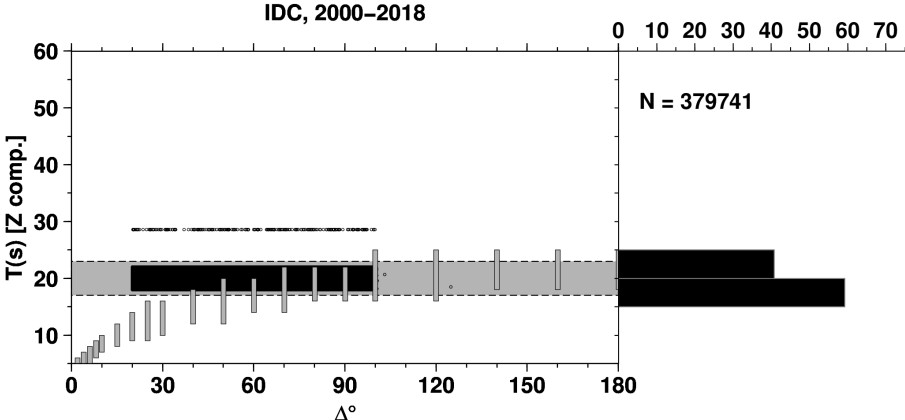

**Figure A4.** As for Fig. 7 but for reporter IDC, vertical component only.

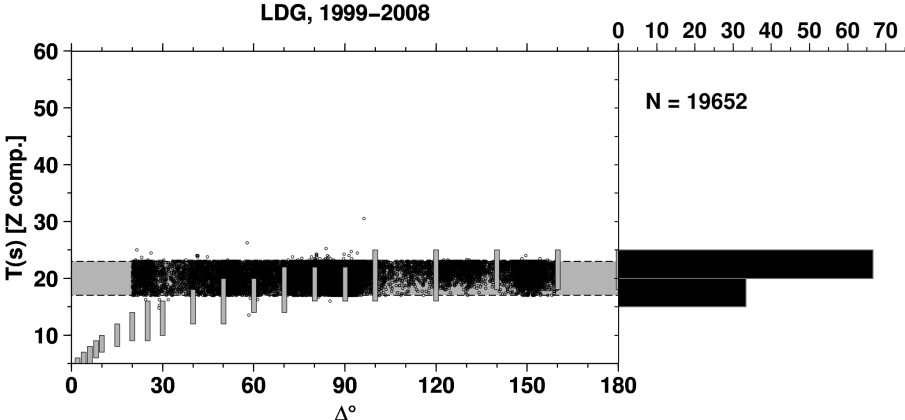

**Figure A5.** As for Fig. 7 but for reporter LDG, vertical component only.



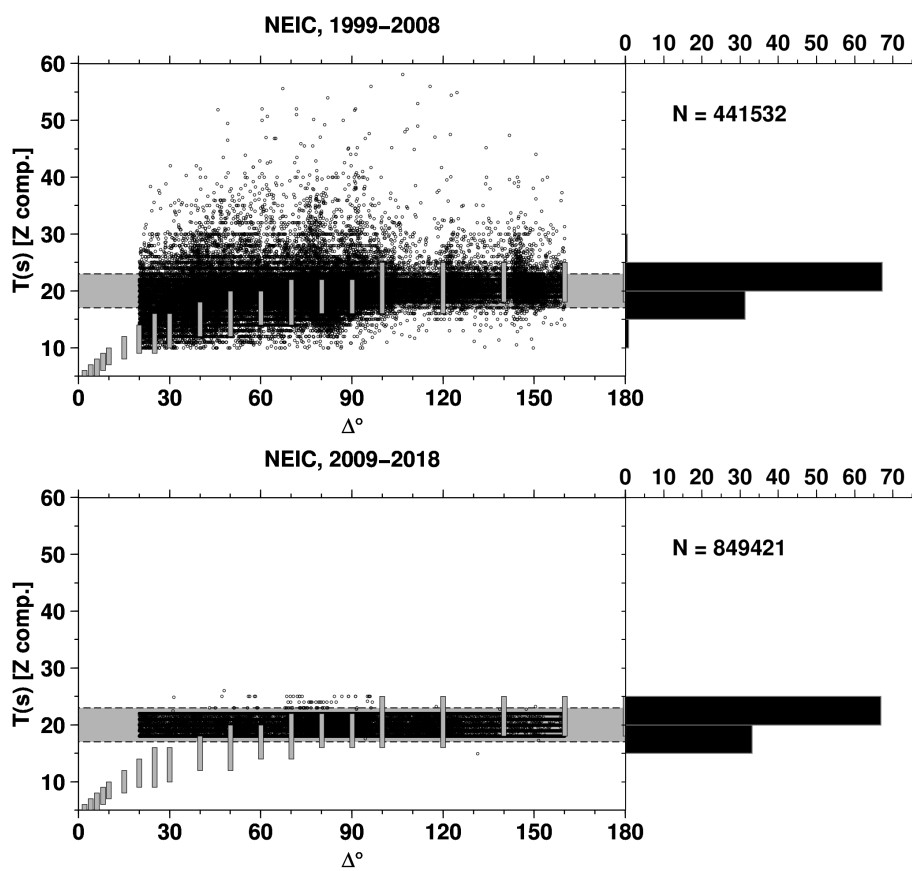

**Figure A6.** As for Fig. 7 but for reporter NEIC, vertical component only. Here we split the plot in two time periods to emphasize the change in 2009 by NEIC in reporting surface wave amplitudes strictly around 20 seconds



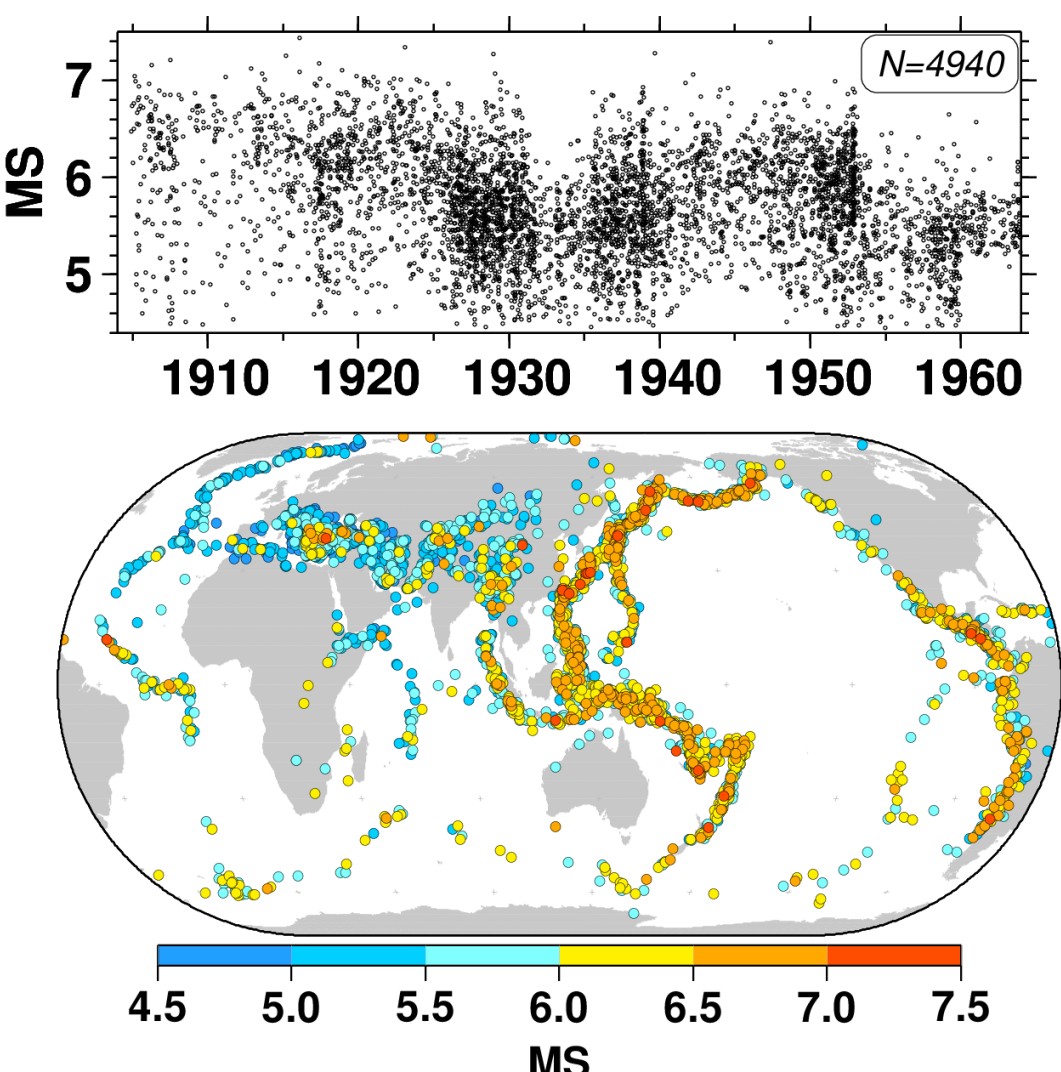

**Figure A7.** Map colour-coded by *MS* and timeline of the pre-1964 earthquakes where, according to our records, *MS* has been computed for the first time. Map drawn using the Generic Mapping Tools (GMT) (Wessel et al., 2013) software.



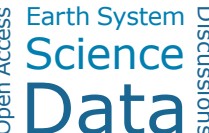

**Table A1.** Examples of $Mw < 8.2$ earthquakes characterized by large differences between $Mw$ and $MS$. $Mw$ is from literature for pre-1976 earthquakes, GCMT otherwise. The last column is the ISC event identifier.

| $Mw \ll MS$ | Origin Time (UTC) | Lat. | Long. | $Mw$ | $MS$ | ISC_evid |
|---|---|---|---|---|---|---|
| | 1906-04-18 13:12:26 | 38.04 | -122.40 | 7.70 | 8.61 | 16957905 |
| | 1915-10-03 06:53:21 | 40.27 | -117.58 | 6.80 | 7.61 | 913944 |
| | 1927-03-07 09:27:41 | 35.56 | 134.99 | 7.00 | 7.82 | 909128 |
| | 1969-07-18 05:24:45 | 38.35 | 119.51 | 6.90 | 7.72 | 807162 |
| | 1970-05-27 19:05:37 | 40.27 | 143.03 | 5.90 | 6.85 | 796053 |
| $Mw \gg MS$ | Origin Time (UTC) | Lat. | Long. | $Mw$ | $MS$ | ISC_evid |
| | 1940-12-28 16:37:44 | 18.20 | 147.46 | 7.70 | 6.60 | 901750 |
| | 1954-08-27 10:54:55 | 23.99 | 143.02 | 7.20 | 6.23 | 891003 |
| | 1983-01-13 09:23:49 | -35.83 | -102.88 | 6.12 | 5.10 | 584568 |
| | 1995-05-26 03:11:15 | 11.89 | 58.02 | 6.46 | 5.42 | 106101 |