# Peer review of "100+ years of recomputed surface wave magnitude of shallow global earthquakes"

_Earth System Science Data, 2021_

## Referee Comment (RC2)

Review of "100+ years of recomputed surface wave magnitude of shallow earthquakes" by Drs. D. Di Giacomo and D. A. Storchak

With the recent advent of high-quality seismic networks, the database of observational seismology has expanded significantly, yet the availability is relatively short, about 3 to 4 decades, and it is desirable to relate the results of modern seismology to those obtained prior to the 1980s. As the authors state, the surface-wave magnitude $M_S$ is among the key parameters that would allow us to compare the modern and old events, thereby allowing us to better understand the long-term seismicity of the Earth.   Unfortunately, the historical data of $M_S$ is incomplete and often confusing, and we encounter many difficulties. This paper describes the results of impressive efforts to establish a more complete historical $M_S$ database, and is a welcome contribution to seismology and merits formal publication in Earth System Science Data.   As the authors state, this is not meant to be a completed product, and future developments are planned and proposed.  Thus, this review consists of some questions on the procedures used, but more importantly, I would like to make some suggestions and caveats with the hope that this database can be made more useful for serious users including myself.

I will make some detailed comments below, and I recommend publication of this manuscript, after the authors consider my comments at their discretion.

Comments and questions on specific points.

Line 25.  Please comment on A and $(A/T)_{max}$.  Is the component specified?   Does $(A/T)_{max}$ literally mean the maximum of A/T,  or can $(A_{max}/T)$ be used as a proxy?

Line 31 to 36. I thought that the basis of Abe's (1981) catalog is Gutenberg's notepad (Goodstein et al., 1980).  Also, I thought that Rothé (1969) is the continuation of Gutenberg and Richter's (1954) Seismicity of the Earth.  As far as I understand, most magnitudes published in Gutenberg and Richter (1954) and Rothé are $M_S$ but some are based on $m_B$.  It appears that Gutenberg's idea of "unified magnitude" had some influence on the magnitudes in these publications.   Some explanations here would be helpful. Richter (1958) would be most useful on this subject.

Line 44. What does "digitize" mean here?  Does it mean to convert printed materials to computer-accessible format?

Line 63. $\sqrt{\left(2*\dfrac{A}{T}\right)^2_{N|E}}$ is an ambiguous  notation.  Please clarify.

Line 70.  Can you elaborate on  "median absolute deviation (SMAD) of the a-trimmed station magnitude"?

Line 198. I agree with the statement "It is of paramount importance that datasets are well-documented and that users know how they are created in order to properly use them for research"

Line 227 to 232. Although I do not have strong objection to the statement here, I think this section reads somewhat strange.  Although "Saturation" and "Under-estimation" can be used in a qualitative

statement, it is important to emphasize here that $M_S$ and Mw are different parameters representing an earthquake "measure" at different periods. Both measures are equally important.

Line 248. I am glad to know that development of the $M_S$ dataset will continue. Although the current database is very useful, for the events before 1970, the number of stations is often very limited with a large azimuthal gap, and it would be extremely helpful to include as many stations as possible. The plans summarized in this manuscript to "digitize" as many more station bulletins as possible are very impressive. Although I can understand the ISC's desire to go back to the original station bulletins, other amplitude data from various secondary sources (e.g., BCIS, Gutenberg notepad (Goodstein et al. 1980), Abe's note (this apparently exists at the Earthquake Research Institute), database used for Rothé (1969), and Lienkaemper (1984)) can be useful, and can be also utilized. Although there could be some small differences in the measuring method of amplitude, period etc in these secondary sources, it seems to me that the biggest uncertainties in the event Ms values come from the limited azimuthal coverage rather than the small differences in the amplitude measurements, and it would be useful to include Ms data from the secondary sources with an appropriate flag, if desired.

Line276 to 292. Conclusions

While I admire the ISC's efforts for establishing a good $M_S$ data base, many investigations have been made using some standard catalogs like Gutenberg and Richter (1953), Richter (1958), Rothé (1969), Duda (1965), Abe (1981), etc. Although there are some differences in details such as the method of amplitude measurements, the method of picking the phase (A vs A/T), attenuation relation (Gutenberg vs. IASPEI formula), the component used, station corrections, and the averaging scheme etc (some of these differences are covered in this manuscript), it would be useful if the authors make some comparisons of magnitude between the new ISC $M_S$ and that from these catalogs. It can be done by some simple figures and by some tables for important events. Again, some of the comparisons may have been already made elsewhere, but it would be useful to show them together in this paper.

There are a few questions myself, and if the authors can provide some insight on them, it will be useful for many serious researchers to fully utilize the ISC catalog.

Questions

The most important material is the "Ms_Dataset" that accompanies this document. Many of the issues raised below are related to the content in "Ms_Dataset".

1) I vaguely remember that Gutenberg and Richter used some station corrections (e.g., Gutenberg, 1944), but I have not seen the list of the station corrections. It is possible that the station corrections include not only the path effects (different attenuation and focusing and defocusing of energy due to multi-pathing), but also some effects of different station practices for measuring the amplitude and applying the instrument gain corrections (static magnification vs. magnification at the period of the waves being measured.)

2) Did any of the stations apply corrections for the depth? It appears that Gutenberg attempted to apply some corrections (Gutenberg, 1944), but I wonder if it is documented somewhere. I am almost certain that excitation of 20s surface waves can be significantly affected by the depth even for the relatively small depth ranges from 0 to 60 km.

3) Are any considerations given to whether the measured waves are Rayleigh type waves or Love type waves. This must have significant effects when $M_S$ from vertical and horizontal components are mixed. Geller and Kanamori (1977) discussed some of the issues.

4) I noticed a few cases in which 2 successive events which are very close in time are given separate $M_S$. It will be helpful if the authors offer some explanations for these cases. Following are just 2 examples related to the 1960 Chilean earthquake.

   (i)     Event 879134 and 879136.

   Event 879134 occurred about 15 min before the $M_S$=8.58 Chilean mainshock (#879136), and is given $M_S$=8.44. Judging from the difference in the amplitude of body waves, it would be very difficult to pull out the surface waves from Event 879134 which are buried in much larger waves of Event 879136.

   (ii)    Event 879127 and 879128

   These events occurred just 2 minutes apart ($M_S$=7.18 for #879127 and $M_S$=7.0 or #879128), and again how the surface waves from these 2 events were separated is unclear.

5) For events after 1990 when the large number of modern global stations were used, a new problem emerged. This topic is closely related to the discussion from line 235 and Table A1. Comparison of $M_S$ and Mw is often very important for understanding the nature of earthquakes. Large differences between $M_S$ and Mw, and $M_S$ between different catalogs can be due to many causes. 1) The real physical characteristics of the event (e.g., slow earthquakes), 2) very limited Mw data (e.g., single measurement), 3) large differences in $M_S$ from different sources.

   Here, the issue is the difference in $M_S$ from different sources. We occasionally see a very large difference ($\Delta M_S > 0.5$) between $M_S$ from different sources. For old events (e.g., before 1970), often the difference is due to a very limited azimuthal coverage. This is to some extent inevitable because simply the number of global stations was relatively small. However, it would be important to assemble as many station Ms data as possible, even if the measurement practice is slightly different at different stations. Even if some measurements do not meet the strict ISC standard, that can be added, with appropriate flags if necessary, to the $M_S$ basic catalog. As I will show later, I think that the variation of $M_S$ with azimnuth is often much larger than that due to the difference in the measuring practice (amplitude, period, attenuation function, etc), and for many research purposes it is important to have a good azimuthal coverage.

   Now going back to modern events (e.g., 1970), we have the luxury of having many and many stations, and often have an opposite problem. Occasionally, we have too many stations in a small azimuthal range with "anomalous" path effects, and this can bias the final station $M_S$ (either some sort of average or median). Since the azimuthal variation of $M_S$ due to the path effect can be as large as $\Delta M_S$=2 unit, this azimuthal bias can produce very confusing results. Of course, this is more of a research subject than catalog-related subject, but it will be helpful if some caveats are given in this paper. I have seen many problematic and questionable cases in the literature, and wish that some more careful discussions are made in this paper so that users are aware of this problem.

Following are some examples from old and modern events.

Event  #878564     1960-03-20     Off east coast of Honshu

The ISC $M_S$ is 7.92, but JMA magnitude $M_{JMA}$ is 7.0.  Although $M_{JMA}$ is not exactly $M_S$, it was calibrated against Gutenberg and Richter's M, and generally believed to be close to $M_S$.   For many Japanese events, $M_{JMA}$ is indeed close to $M_S$ (e.g., Utsu, 2001, Relationships between magnitude scales).  Thus, this large difference caught my attention.  Figure 1 compares the azimuthal variation of $M_S$ taken from "Ms_Dataset" and that computed for a nearby Mw=7.3 event (3/9/2011) using the global data.  The azimuthal variation patterns of $M_S$ for the 2 events are very similar, and large. The range is almost 2 magnitude unit (6.5 to 8.5) for the 1960 Sanriku event, and 1.5 unit for the 2011 event.  Since the number of stations for the 2011 event is very large (what are shown on the figure represent only a subset), and the azimuthal coverage is uniform, the median appears to be well defined. On the other hand, for the 1960 event, the data set is dominated by the stations in the azimuthal range from 300 to 360 deg.   Although there is some evidence that the 1960 event was a slow earthquake, it should not affect $M_S$, and the users should be aware of this strong azimuthal variation of $M_S$.

[Figure]

19600320 Sanriku, $M_S$_ISC=7.92, $M_{JMA}$=7.0          20110309  Tohoku, $M_{S\_median}$=7.3

Fig. 1

Figure 2 is an example given in Table A1 of this manuscript: the 3/7/1927 event with an ISC $M_S$ of 7.82.  To compare this event with a recent event in the same area, the $M_S$ data of the 2016 Tottori earthquake ($M_S$=6.2) are shown.  Again the 1927 case is strongly influenced by the stations in the azimuth range of 300° to 360°. A somewhat similar pattern is seen for the 2016 Tottori earthquake, but since the azimuthal coverage is uniform, the median ($M_S$=6.2) seems to be fairly stable, and in fact this value agrees well with the value quoted in GCMT catalog .

[Figure]

19270307 Tango MS_ISC=7.82, $M_{JMA}$=7.3

[Figure]

20161021 Tottori, $M_{S\_median}$=6.2

Fig. 2

Figure 3 shows an example of the 2002 Denali earthquake (Mw=7.8). The $M_S$ value given by NEIC and listed in the GCMT catalog is 8.5, presumably an average of more station data than those shown in Figure 3. The existing global stations are far more than those shown in Figure 3, especially in the azimuth of about 100°, and an average can become very large. I do not know exactly what an averaging scheme is used. In this particular case, if we take a bin-average, we get the following:

| Azimuth range | number of stations | bin average |
|---|---|---|
| 0-45 | 20 | 7.536 |
| 45-90 | 4 | 8.468 |
| 90-135 | 14 | 8.539 |
| 135-180 | 8 | 8.273 |

The average of the bin average is 7.85, very close to the ISC value of 7.82. This is a new problem with recent events with so many stations, and the averaging scheme is very important.

[Figure]

20021103 Denali MS_ISC=7.82, $M_{S\_GCMT}$=8.6

[Figure]

Fig. 3

This bring us to a problem with old events again. As discussed on line 238 and illustrated in Table A1 of this manuscript, the famous 1906 San Francisco is given $M_{S\_ISC}$=8.61. However, as shown in Figure 4, only 6 stations are used, with 4 stations essentially in the same azimuth, and it is hard to assess the uncertainty of the assigned $M_S$ value. Since the Gutenberg notepad lists station $M_S$ values from some 16 stations, and Gutenberg and Richter (1954) gave MS= 8 ¼. I like this notation which probably implies an uncertainty of ¼ magnitude unit. Unfortunately, many journals demand that it should be written as 8.25 which has a very different implication for the uncertainty. Also, I suspect that Gutenberg and Richter's assignment of "quality, A, B, and C, or a, b, and c is somewhat subjective on the basis of their experience, but in case of this kind of data to which rigorous statistical method is hard to apply, I believe that it is a very reasonable practice. Actually, Lienkaemper (1984) examined Gutenberg's notepad data, and came up with M=8.3. Although I did not follow exactly what he did, he did use the 16 stations. I suspect that the data listed in the Gutenberg's note pad did not meet the strict criterion of ISC, and were not adopted in "$M_S$_Dataset" (I may be wrong.). However, as shown in the examples presented in Figures 1 to 4 in this review, the azimuthal station coverage is so important for obtaining a reasonable average that I would like to see as many station data as possible in the ISC catalog, even if the measurement procedure was slightly different. Overall, my experience is that the difference caused by the limited azimuthal coverage is far greater than that caused by the difference in the station practice.

190604  San Francisco  $M_{S\_ISC}=8.61$, $M_{\_GR}=8\frac{1}{4}$, $M_{LK}=8.3$

[Figure]

Fig. 4

My comments above are in no way the criticism of the ISC practice and catalog, but they represent my hope that the ISC catalogs will be used most effectively and properly by serious users.  The historical data are important but they can have all kinds of problems and uncertainties, and very often rigorous handling is not possible.  Nevertheless, these data do contain historical information which we cannot get otherwise.  After all, how to use the data base and interpret it is ultimately the responsibility of the users, rather than the catalog producers, but it is most important that the catalog producers provide adequate caveats to the users so that the catalogs can be carefully used for understanding the Earth's seismicity.

---

## Referee Comment (RC3)

**COMMENTS**

on the manuscript entitled:

*"100+ years of recomputed surface wave magnitude of shallow earthquakes"*
by D. Di Giacomo and D.A. Storchak

submitted for publication in "Earth System Science Data (ESSD)"

**A.  General Comment**

Earthquake catalogs, extending over a wide period and covering the globe are useful tools for many studies. Two are the critical preconditions that must be fulfilled: accuracy in their focal parameters and homogeneity regarding the scale in which their magnitudes are expressed. Considering the fact that it is not suffering saturation but only at its large values, Ms is a suitable magnitude for such studies.

In this spirit, I believe that this work is very useful and it is my sense that its outcome (the catalog) is going to be extensively used in the future.

The paper is well written and its content corresponds to its title. There are some minor issues that I will describe below, which, if clarified, I believe will further improve the manuscript.

Concluding, it is my opinion that the manuscript can be accepted for publication after some minor revision.

Following are my comments in details.

**B.  Specific Comments**

1) In the 1st paragraph of "Introduction" the basic pros of surface wave magnitude, Ms, are mentioned. I believe that the cons (e.g. inability of Ms estimation by using records of short period instruments and, therefore, of small local earthquakes, possible underestimation for very strong earthquakes) should be mentioned too.

2) Page 3: In the square root, the factor "2" must be out of the brackets:

$$\left(\frac{A}{T}\right)_H = \sqrt{2 \cdot \left(\frac{A}{T}\right)^2_{N/E}}$$

3) How have you estimated the final surface wave magnitude if more than one Ms values were available? Mean value? Weighted mean? Have you applied any filters to avoid contamination that could be caused by one or more potentially incorrect magnitude values that may deviate significantly from the majority of the rest?

4) The final catalog includes only events with recomputed Ms magnitudes, meaning that this catalog is not complete, as it possibly misses earthquakes which could be included in catalogs published by other authors, covering wide regions and extending over wide time periods, but with magnitudes consistent (not original) to the standard Ms, (e.g. Karnik 1996).

5) In lines 72-74 you mention that: *"The locations adopted in this work come from the ISC-GEM Catalogue (Bondár et al., 2015; Di Giacomo et al., 2018) between 1904 and 1963 and the rebuilt ISC Bulletin (Storchak et al., 2017, 2020) from 1964 onward".* Looking in the ISC-GEM catalog I could not find some earthquakes included in your catalog. Indicatively I mention the following events: 1904-12-02, 02:19:12; 1904-12-11, 17:05:42; 1908-01-31, 04:49:15 etc. These events are also not included in the online ISC bulletins. Figure 3 clearly shows that data before ~1950 are coming exclusively from ISC. So, which is their origin?

6) Chapter 4 entitled "Catalogue Properties" gives a detailed and very useful analysis of the time-history of Ms scale. However, it looks there are four gaps regarding earthquakes of Ms<6.0 that can be observed in figure 10: one at ~1920, the second during 1940-1950, the third between 1960 and 1978 and the fourth between 1980 and 1984. The authors are right about the impact of World War 2, which justifies the second gap. According to the authors (see chapter 6), there is further work to be done that will possibly allow some of the above gaps to be covered. So, consider this as just a remark.

7) In the same chapter (4) and in lines 160-170 there is an analysis of the features of the formed catalog. It is mentioned there how the completeness magnitude, Mc, is distributed over time. There is a point here that, in my opinion, needs clarification. To proceed to a meaningful Mc estimation and to study its variation with time it is necessary to know first if there are earthquakes systematically missing from the data set. I mean, are there any earthquakes whose focal parameters are known but they are not included in the catalog because it was not possible to have Ms estimation for them? If yes, then I believe that the term "completeness magnitude" should be avoided as, at least literally, it has another meaning.

8) Figure 11 & Lines 180-190 (a follow up of the previous comment): The rates shown in figure 11 do not necessarily show variation of completeness magnitudes over time.

9) Figure 12: The Ms underestimation for very strong earthquakes (e.g. M$\geq$8.0) has been already observed and noticed (e.g. Heaton et al., 1986). For the example of the magnitude of the Aleutian earthquake of April 11, 1946, the magnitude reports in the ISC bulletin are: Ms=7.3 (after Abe, "Phys. Earth planet. Interiors", 1981); Ms=7.1 & Mw=8.0 (after Pacheco & Sykes, "Bull. Seism. Soc. Am.", 1992); Mw=8.6 (after López & Okal, "Geophys. J. Int.", 2006). These values are clearly showing (as the authors of this manuscript state) that Ms values underestimated the real magnitude of this great event. Therefore, although the data in figure 12 are not many, it can be stated that the "saturation" of Ms for values over ~8.0 is confirmed here and should therefore be considered as a fact.

**C. Technical Corrections**

I could not make it to locate in the text the following two references (included in the "References"):

1) Line 339: Bormann (2012)
2) Line 350: Di Giacomo and Storchak (2016)

---

## Community Comment (CC1)

In the line 69

$$\left(\tfrac{A}{T}\right)_H = \sqrt{\left(2 * \tfrac{A}{T}\right)^2_{N|E}}.$$   must be   $$\left(\tfrac{A}{T}\right)_H = \sqrt{2 * \left(\tfrac{A}{T}\right)^2_{N|E}}.$$

Value is too large as it is in the case of $(A/T)_N=(A/T)_E$.

Another concern is if only one component was reported, it might be possible that the larger value of $A/T_N$ or $A/T_E$ was chosen in the report. In the early days of the observation in Japan, stations reported the amplitudes of both horizontal components, but the central office chose only one of the larger component to fill out the central report for labor saving.

The JMA magnitude is determined by the Tsuboi's formula (1959) as

$M_{JMA}=1.73\log \Delta +\log\sqrt{(A_N{}^2+A_E{}^2)}-0.83.$

Taking into account situation of the observation Utsu(1979) applied $(\sqrt{2}-0.05)*\log(Amax)$ instead of $\sqrt{2}*\log(Amax)$ for magnitude determination of earthquakes in the early period of observation in his study.

Based on the comparison of $\sqrt{(A_N{}^2+A_E{}^2)}$ with $\sqrt{A_{Max}{}^2}$ for about thousand cases, Hamada et al. (2001) adopted 1.25 instead of $\sqrt{2}$ for their study. These differences are at most 0.1 in M, but I think they are worth commenting on here.

Sorry for the following my references are all in Japanese.
Tsuboi C. (1959) Determination of the GUTENBERG-RICHTER'E Magnitude of Earthquakes occurring in and near Japan, Zisin, 2, Vol 7, 185-193.
https://www.jstage.jst.go.jp/article/zisin1948/7/3/7_3_185/_pdf

Ustu T.,1979, Seismicity of Japan from 1885 through 1925 : A New Catalog of Earthquakes of M=6 Felt in Japan and Smaller Earthquakes Which Caused Damage in Japan, Bul.Earth.Res.Inst.,Vol54, 253-308.
https://repository.dl.itc.u-tokyo.ac.jp/record/33106/files/ji0542002.pdf

Hamada N. et al. (2001) A Comprehensive Study of Aftershocks of the 1923 Kanto Earthquake, Zisin, 2, Vol 54, 251-265.
https://www.jstage.jst.go.jp/article/zisin1948/54/2/54_2_251/_pdf

---

## Author Comment (AC1)

**Author's Response to Nobuo Hamada (CC1) (PAPER: https://doi.org/10.5194/essd-2021-266)**

Domenico Di Giacomo & Dmitry A. Storchak

November 2021

We thank the community comment by Nobuo Hamada on our submission. Below we reply in detail to each point showing the Community Comment (CC) in bold and the Author Response (AR) in italic.

**CC:**

**In the line 69**

$$\left(\tfrac{A}{T}\right)_H = \sqrt{\left(2*\tfrac{A}{T}\right)^2_{N|E}}\cdot$$ **must be** $$\left(\tfrac{A}{T}\right)_H = \sqrt{2*\left(\tfrac{A}{T}\right)^2_{N|E}}\cdot$$

**Value is too large as it is in the case of $(A/T)_N=(A/T)_E$.**

*We thank the CC for pointing out the typeset error (he was the first to spot it soon after discussion started).*

**Another concern is if only one component was reported, it might be possible that the larger value of $A/T_N$ or $A/T_E$ was chosen in the report. In the early days of the observation in Japan, stations reported the amplitudes of both horizontal components, but the central office chose only one of the larger component to fill out the central report for labor saving.**

*Japanese stations are a minor contributor to MS for our global dataset. We are aware that single component measurements are not ideal, but it is better than no measurement at all. We agree with reviewer #2 that to improve this dataset we need some degree of flexibility in the data we allow in our procedure for computing MS (at least up to the 1960s). Hence, we ought to include single component measurements.*

**The JMA magnitude is determined by the Tsuboi's formula (1959) as $M_{JMA}=1.73\log\Delta+\log\sqrt{(A_N^2+A_E^2)}-0.83$. Taking into account situation of the observation Utsu(1979) applied $(\sqrt{2}-0.05)*\log(Amax)$ instead of $\sqrt{2}*\log(Amax)$ for magnitude determination of earthquakes in the early period of observation in his study. Based on the comparison of $\sqrt{(A_N^2+A_E^2)}$ with $\sqrt{A_{Max}^2}$ for about thousand cases, Hamada et al. (2001) adopted 1.25 instead of $\sqrt{2}$ for their study. These differences are at most 0.1 in M, but I think they are worth commenting on here.**

*We adopt widely used standards for computing MS and the issues raised by the CC are certainly worth investigating but more for a regional study (as is the case for JMA magnitude), and we encourage users of our dataset to do that.*

[revised manuscript text omitted]

---

## Author Comment (AC2)

**Author's Response to Kenji Satake (RC1) (PAPER: https://doi.org/10.5194/essd-2021-266)**

Domenico Di Giacomo & Dmitry A. Storchak

November 2021

We thank the reviewer #1, Kenji Satake, for his comments and suggestions. Below we reply in detail to each point showing the Referee Comments (RC) in bold and the Author Response (AR) in italic. All RC1 suggestions have been accepted and included in the revised version of the manuscript (annotated manuscript after our answers).

**RC:**

**Surface wave magnitude is the only scale to measure global earthquake size for more than a century. Currently, seismic moment is considered to be the best parameter to quantify the earthquake size, and moment magnitude Mw, another way to express seismic moment, is used for recent earthquakes. To calculate seismic moment or Mw, seismic waveform modeling is required, but such waveforms are systematically available only after 1960's. On the other hand, magnitude scales (including MS) are based on reported values of amplitude and period, that are available for more than a century. The authors have digitized the old seismological station bulletins (reports of arrival times, amplitudes and periods of seismograms) at the International Seismological Centre, and made MS catalog since 1904 for global earthquakes.**

**The paper is basically well written and almost publishable as is in Earth System Science Data. I provide some comments, which the authors may want to consider for making final manuscript.**

- **Title: a keyword "global" may be added to express the global distribution of earthquakes.**

*We have added the word "global" in the title.*

- **Introduction well summarize the Ms scale and previous catalogs. In line 42, "over 46,000 earthquakes with MS ≥4.5" may be somewhat misleading, as Figure 10 shows that Mc is much larger in the early period.**

*We have updated the sentence as "we present a revised MS catalogue (cut-off magnitude of 4.5) listing over 46,000 earthquakes as well as the underlying station data"*

- **Before Section 2 (Recomputing Ms), moving some parts of Section 3 (Station data) on collection of station bulletin at ISC, i.e., the first three paragraphs (up to line 100) or seven paragraphs (up to line 118), would be beneficial for readers. Such reorganization would require renumbering of figures, but solve appearance of "reporters" in line 57 before defined as "hereafter also referred as reports or data contributors" in line 89.**

*We have accepted the reviewer suggestion and moved the first three paragraphs of Section 3 "Station data" to the beginning of section 2 now entitled "Reporters and MS recomputation".*

- **Line 120: "secondary gap" may need more explanation, e.g., "the largest azimuthal gap in which only one station exists, and the error of this station may bias the solution". It may be also worth mentioning that unlike body waves which radiates three- dimensionally and the stations are ideally distributed in the focal sphere, station coverage for surface waves are evaluated only in azimuthal direction, and radiation patters of surface waves are symmetric (either two-lobed or four-lobed).**

*We have included the reviewer suggestion by updating the sentence as "secondary gap (i.e., the largest azimuthal gap in which only one station exists, and the quality of the data at that station may bias the solution)". Regarding the second suggestion, after the citation "Von Seggern (1970)" we have added "although the latter is symmetric for surface waves (either two-lobed or four-lobed".*

- **Figure 10 is one of the important results of this paper. While the authors attributed the gap of small earthquakes (M< 5.5) between 1940 and 1950 to the World War II, similar (actually wider and clearer) gap is seen in magnitude timeline (bottom figure) between 1960 and 1979. Any explanation of this gap ?**

*We have added a discussion on the period 1960-1977 by adding the following:*

*"The period 1960-1977 also features less earthquakes below 5.5 than previous and following decades. This is due both to the limited number of stations available and the fact that we digitized surface wave data from the 1960s printed station bulletins only for earthquakes selected in the first version of the ISC-GEM Catalogue (magnitude 5.5 and above, Storchak et al., 2013). In Section 6 we propose activities that are likely to mitigate significantly the deficiencies of the ISC MS dataset in most of the 1960s-1970s."*

- **"Saturation" issue (Figure 12) is also important. The authors mention that variation in MS saturation (difference between Mw and MS) is larger for earthquakes 8<Mw<9 than those with Mw>9. If we ignore 1946 Aleutian (typical tsunami earthquake) and 1952 Kamchatka (GR and Abe gave larger MS values), the difference may be within the scatter of smaller magnitudes. Incidentally, Maule earthquake is 2010, not 2018 (line 219 typo), and neither 1965 Rat Island or 2005 Nias earthquakes is considered as "tsunami earthquake"**

*We thank the reviewer for pointing out the typo in the year of the Maule earthquake (2010 instead of 2018). We also replaced the sentences*

[revised manuscript text omitted]

---

## Author Comment (AC3)

**Author's Response to Anonymous Reviewer (RC2) (PAPER: https://doi.org/10.5194/essd-2021-266)**

Domenico Di Giacomo & Dmitry A. Storchak

November 2021

We thank the reviewer #2 for the comments and suggestions. Below we reply in detail to each point showing the Referee Comments (RC) in bold and the Author Response (AR) in italic. Most of the RC2 suggestions have been accepted and included in the revised version of the manuscript (annotated manuscript after our answers).

**RC:**
**With the recent advent of high-quality seismic networks, the database of observational seismology has expanded significantly, yet the availability is relatively short, about 3 to 4 decades, and it is desirable to relate the results of modern seismology to those obtained prior to the 1980s. As the authors state, the surface-wave magnitude MS is among the key parameters that would allow us to compare the modern and old events, thereby allowing us to better understand the long-term seismicity of the Earth. Unfortunately, the historical data of MS is incomplete and often confusing, and we encounter many difficulties. This paper describes the results of impressive efforts to establish a more complete historical MS database, and is a welcome contribution to seismology and merits formal publication in Earth System Science Data. As the authors state, this is not meant to be a completed product, and future developments are planned and proposed. Thus, this review consists of some questions on the procedures used, but more importantly, I would like to make some suggestions and caveats with the hope that this database can be made more useful for serious users including myself.**

*We thank the reviewer for this general comment.*

**I will make some detailed comments below, and I recommend publication of this manuscript, after the authors consider my comments at their discretion.**

**Comments and questions on specific points.**

**Line 25. Please comment on A and (A/T)max. Is the component specified? Does (A/T)max literally mean the maximum of A/T, or can (Amax/T) be used as a proxy?**

*We do not feel to change the text in the Introduction because there we simply report the original definition by the creators of the MS Moscow-Prague formula. However, in the Section 2 the use of the three components is detailed and we have added the following sentences to address the reviewer questions regarding Amax/T versus (A/T)max:*

*"Although our procedure finds the maximum of A/T within the reading, a reporter may have provided single component measurements of Amax/T".*

**Line 31 to 36. I thought that the basis of Abe's (1981) catalog is Gutenberg's notepad (Goodstein et al., 1980). Also, I thought that Rothé (1969) is the continuation of Gutenberg and Richter's (1954) Seismicity of the Earth. As far as I understand, most magnitudes published in Gutenberg and Richter (1954) and Rothé are MS but some are based on mB. It appears that Gutenberg's idea of "unified magnitude" had some influence on the magnitudes in these publications. Some explanations here would be helpful. Richter (1958) would be most useful on this subject.**

*We feel that it is not the case for our manuscript to debrief the reader on the history of past earthquake catalogues. Interested readers can look in the references of the papers already cited.*

**Line 44. What does "digitize" mean here? Does it mean to convert printed materials to computer- accessible format?**

*We have added "(i.e., converted from printed to computer accessible format)" after "digitize".*

**Line63. $\sqrt{\left(2*\dfrac{A}{T}\right)^2_{N|E}}$ is an ambiguous notation. Please clarify.**

*As pointed by other reviewers, we thank the reviewer #2 for pointing out the typeset error.*

**Line 70. Can you elaborate on "median absolute deviation (SMAD) of the a-trimmed station magnitude"?**

*From the sorted station magnitude distribution, the median is computed and the SMAD is obtained by trimming 20% the bottom and top end of the distribution. This practice has been used for several years in ISC procedures and it is well-documented, hence we did not add further explanations in the text.*

**Line 198. I agree with the statement "It is of paramount importance that datasets are well-documented and that users know how they are created in order to properly use them for research"**

*Thanks.*

**Line 227 to 232. Although I do not have strong objection to the statement here, I think this section reads somewhat strange. Although "Saturation" and "Under-estimation" can be used in a qualitative statement, it is important to emphasize here that MS and Mw are different parameters representing an earthquake "measure" at different periods. Both measures are equally important.**

*We agree with the reviewer suggestion to emphasize the importance of both magnitude types. Hence, we added in the text "We also suggest that Ms and Mw, as expressions at different periods of the earthquake size, should be used together to better characterize an earthquake the source properties of an earthquake."*

**Line 248. I am glad to know that development of the MS dataset will continue. Although the current database is very useful, for the events before 1970, the number of stations is often very limited with a large azimuthal gap, and it would be extremely helpful to include as many stations as possible. The plans summarized in this manuscript to "digitize" as many more station bulletins as possible are very impressive. Although I can understand the ISC's desire to go back to the original station bulletins, other amplitude data from various secondary sources (e.g., BCIS, Gutenberg notepad (Goodstein et al. 1980), Abe's note (this apparently exists at the Earthquake Research Institute), database used for Rothé (1969), and Lienkaemper (1984)) can be useful, and can be also utilized. Although there could be some small differences in the measuring method of amplitude, period etc in these secondary sources, it seems to me that the biggest uncertainties in the event Ms values come from the limited azimuthal coverage rather than the small differences in the amplitude measurements, and it would be useful to include Ms data from the secondary sources with an appropriate flag, if desired.**

*As outlined in Section 6, we are eager to improve the dataset by adding as much data as possible. We would be happy explore new sources and help from the community is welcome.*

**Line276 to 292. Conclusions**

**While I admire the ISC's efforts for establishing a good MS data base, many investigations have been made using some standard catalogs like Gutenberg and Richter (1953), Richter (1958), Rothé (1969), Duda (1965), Abe (1981), etc. Although there are some differences in details such as the method of amplitude measurements, the method of picking the phase (A vs A/T), attenuation relation (Gutenberg vs. IASPEI formula), the component used, station corrections, and the averaging scheme etc (some of these differences are covered in this manuscript), it would be useful if the authors make some comparisons of magnitude between the new ISC MS and that from these catalogs. It can be done by some simple figures and by some tables for important events. Again, some of the comparisons may have been already made elsewhere, but it would be useful to show them together in this paper.**

*We avoided to add magnitude comparisons with previous catalogues as a large literature is available to this regard. We point out in the text that Abe's catalogue is of very good quality and that magnitude comparison with our recomputed Ms are available in Di Giacomo et al. (PEPI 2015). Since Abe's catalogue is mostly for MS 6.5 and above, we feel that we would not add much by updating that figure. Nevertheless, the biggest advance of our MS dataset consists in the thousands of earthquakes for which we provide for the first time MS, and those cannot be compared to any other catalogue.*

**There are a few questions myself, and if the authors can provide some insight on them, it will be useful for many serious researchers to fully utilize the ISC catalog.**

**Questions**

**The most important material is the "Ms_Dataset" that accompanies this document. Many of the issues raised below are related to the content in "Ms_Dataset".**

1. **I vaguely remember that Gutenberg and Richter used some station corrections (e.g., Gutenberg, 1944), but I have not seen the list of the station corrections. It is possible that the station corrections include not only the path effects (different attenuation and focusing and defocusing of energy due to multi-pathing), but also some effects of different station practices for measuring the amplitude and applying the instrument gain corrections (static magnification vs. magnification at the period of the waves being measured.)**

*To the best of our knowledge, magnitude station corrections are not implemented in any global agency that provide magnitude determination from stations worldwide. However, that does not mean that users can analyse our dataset and obtain station correction terms. These, however, are not part of our submission.*

2. **Did any of the stations apply corrections for the depth? It appears that Gutenberg attempted to apply some corrections (Gutenberg, 1944), but I wonder if it is documented somewhere. I am almost certain that excitation of 20s surface waves can be significantly affected by the depth even for the relatively small depth ranges from 0 to 60 km.**

*The literature covers this aspect of MS. Recently, Petrova and Gabsatarova (J. Seism. 2020, and references therein can be found for the literature on the subject) investigated depth corrections to the Moscow MS. It is true that surface wave excitation gets smaller with earthquake depth, hence standard procedures limit MS computation to 60 km (IASPEI, 2013). Depth effect up to 60 can be indeed studied and, again, we encourage users to analyse our dataset in this respect.*

3. **Are any considerations given to whether the measured waves are Rayleigh type waves or Love type waves. This must have significant effects when MS from vertical and horizontal components are mixed. Geller and Kanamori (1977) discussed some of the issues.**

*MSZ (Rayleigh waves) and MSH (Love and Raileigh waves) values are available in our dataset. The use of horizontal components is necessary especially in the first ~70 years of the last century due to the scarcity of vertical component instruments worldwide. In recent decades the vertical component is the most reported to the ISC. We generally find good agreement between MSZ and MSH although discrepancies are expected. As in previous replies, the dataset provides*

*the full information behind a MS network value, and users have the possibility to analyse in many ways our dataset.*

4. **I noticed a few cases in which 2 successive events which are very close in time are given separate MS. It will be helpful if the authors offer some explanations for these cases. Following are just 2 examples related to the 1960 Chilean earthquake.**
    **(i) Event 879134 and 879136.**
       **Event 879134 occurred about 15 min before the MS=8.58 Chilean mainshock (#879136), and is given MS=8.44. Judging from the difference in the amplitude of body waves, it would be very difficult to pull out the surface waves from Event 879134 which are buried in much larger waves of Event 879136.**

*We have verified the association of all reading to both events and we confirm that we associated the surface waves to each event as reported in the original bulletins. The only exception to that is the CLL reading that should be reassociated to evid 879136 (its reassociation, however, will not change significantly the MS for both earthquakes).*
*The Soviet Union stations are the most important contributors for both events and their original reports can be found at PDF page 134 of*
[http://www.isc.ac.uk/printedStnBulletins/Bulletins_scans/URSS/Moscow/](http://www.isc.ac.uk/printedStnBulletins/Bulletins_scans/URSS/Moscow/)
[Seism_Bull_1960_AcademyofSciences_URSS.pdf](http://www.isc.ac.uk/printedStnBulletins/Bulletins_scans/URSS/Moscow/Seism_Bull_1960_AcademyofSciences_URSS.pdf).
*The only station currently available for both events is Riverview (station RIV). At page 34 of its original bulletin at*
[http://www.isc.ac.uk/printedStnBulletins/Bulletins_scans/Australia/Riverview/](http://www.isc.ac.uk/printedStnBulletins/Bulletins_scans/Australia/Riverview/)
[Seism_Bull_1960-1961_Riverview.pdf](http://www.isc.ac.uk/printedStnBulletins/Bulletins_scans/Australia/Riverview/Seism_Bull_1960-1961_Riverview.pdf) *it is possible to find the readings for both earthquakes.*
*As the two earthquakes are co-located and separated by 15 minutes, it is possible to identify the surface waves at a given site for both events. However, we agree with the reviewer that it is a challenging situation to get a reliable MS in such cases as the surface wave trains can be mixed, and depending on the relative size of the pair of earthquakes it can be very difficult to properly identify and measure the surface waves belonging to each event. For RIV case, however, the surface wave phases M are separated by about 12-13 minutes between the two events, which is in line with what we would expect in such a case. Unfortunately, no time is given in the Soviet Union bulletin for the surface waves. As we explain in the text, we plan to top up the station magnitude contribution in the 1960s and the 1960 Chilean sequence will likely benefit from that activity, and it may lead us to revise and improve the MS solution for both earthquakes.*

    **(ii) Event 879127 and 879128**
       **These events occurred just 2 minutes apart (Ms=7.18 for #879127 and MS=7.0 or #879128), and again how the surface waves from these 2 events were separated is unclear.**

*The same arguments apply to this pair.*

5. **For events after 1990 when the large number of modern global stations were used, a new problem emerged. This topic is closely related to the**

**discussion from line 235 and Table A1. Comparison of MS and Mw is often very important for understanding the nature of earthquakes. Large differences between MS and Mw, and MS between different catalogs can be due to many causes. 1) The real physical characteristics of the event (e.g., slow earthquakes), 2) very limited Mw data (e.g., single measurement), 3) large differences in MS from different sources.**

*We generally agree with these comments and, although is not our aim in this manuscript, we remind the reader that difference ought to occur and are likely to give investigators cases worth studying.*

**Here, the issue is the difference in MS from different sources. We occasionally see a very large difference (ΔMS > 0.5) between MS from different sources. For old events (e.g., before 1970), often the difference is due to a very limited azimuthal coverage. This is to some extent inevitable because simply the number of global stations was relatively small. However, it would be important to assemble as many station Ms data as possible, even if the measurement practice is slightly different at different stations. Even if some measurements do not meet the strict ISC standard, that can be added, with appropriate flags if necessary, to the MS basic catalog. As I will show later, I think that the variation of MS with azimnuth is often much larger than that due to the difference in the measuring practice (amplitude, period, attenuation function, etc), and for many research purposes it is important to have a good azimuthal coverage.**

*We cannot agree more with the reviewer that it is important to gather as much data as possible and be flexible with the restriction criteria. Indeed, we detail how we adopted an expanded procedure for MS calculation before 1964. We are aware of the azimuthal limitation in the first 70 years and for the first time in a catalogue of this kind we provide the azimuthal gaps in the catalogue file for station magnitudes and discussed the matter with Figure 6.*

**Now going back to modern events (e.g., 1970), we have the luxury of having many and many stations, and often have an opposite problem. Occasionally, we have too many stations in a small azimuthal range with "anomalous" path effects, and this can bias the final station MS (either some sort of average or median). Since the azimuthal variation of MS due to the path effect can be as large as ΔMS=2 unit, this azimuthal bias can produce very confusing results. Of course, this is more of a research subject than catalog-related subject, but it will be helpful if some caveats are given in this paper. I have seen many problematic and questionable cases in the literature, and wish that some more careful discussions are made in this paper so that users are aware of this problem.**

*As mentioned in previous reply, we provide users with the full picture of the data contributing to the network MS. Hence, cases with limited azimuthal coverage are easily identified. We agree that MS could be biased to some extent in such cases. Again, tough, we have included the azimuthal gaps in the dataset and briefly discussed the implications of large azimuthal gaps. Nevertheless, we wish to satisfy the reviewer desire to stress more this issue and included in the manuscript the case*

*of the 1960-03-20 off east coast of Honshu (evid = 878564), see new Figure A1 in the revised manuscript and related text in Section 3.*

**Following are some examples from old and modern events.**

**Event #878564 1960-03-20 Off east coast of Honshu**

**The ISC MS is 7.92, but JMA magnitude MJMA is 7.0. Although MJMA is not exactly MS, it was calibrated against Gutenberg and Richter's M, and generally believed to be close to MS. For many Japanese events, MJMA is indeed close to MS (e.g., Utsu, 2001, Relationships between magnitude scales). Thus, this large difference caught my attention. Figure 1 compares the azimuthal variation of MS taken from "Ms_Dataset" and that computed for a nearby Mw=7.3 event (3/9/2011) using the global data. The azimuthal variation patterns of MS for the 2 events are very similar, and large. The range is almost 2 magnitude unit (6.5 to 8.5) for the 1960 Sanriku event, and 1.5 unit for the 2011 event. Since the number of stations for the 2011 event is very large (what are shown on the figure represent only a subset), and the azimuthal coverage is uniform, the median appears to be well defined. On the other hand, for the 1960 event, the data set is dominated by the stations in the azimuthal range from 300 to 360 deg. Although there is some evidence that the 1960 event was a slow earthquake, it should not affect MS, and the users should be aware of this strong azimuthal variation of MS.**

[Figure]

[Figure]

Fig. 1

*We are aware of the large MS of this event compared to other magnitudes, including MJMA. As mentioned above, we have included this example in the text (Section 3) to stress even more the limitation due to the azimuthal distribution of the stations contributing to the network MS. We consider this the best example in this respect among the ones highlighted by the reviewer. As the event is in 1960, we hope to be able to add station magnitudes in different azimuths and update accordingly the MS for this event in future versions of the dataset.*

**Figure 2 is an example given in Table A1 of this manuscript: the 3/7/1927 event with an ISC MS of 7.82. To compare this event with a recent event in the same area, the MS data of the 2016 Tottori earthquake (MS=6.2) are shown. Again the 1927 case is strongly influenced by the stations in the azimuth range**

**of 300° to 360°. A somewhat similar pattern is seen for the 2016 Tottori earthquake, but since the azimuthal coverage is uniform, the median (MS=6.2) seems to be fairly stable, and in fact this value agrees well with the value quoted in GCMT catalog .**

[Figure]

Fig. 2

*As the event is included in Table A1, we do not think it is necessary to discussed it any further. Points regarding the azimuthal issue are remarked in the revised manuscript.*

**Figure 3 shows an example of the 2002 Denali earthquake (Mw=7.8). The MS value given by NEIC and listed in the GCMT catalog is 8.5, presumably an average of more station data than those shown in Figure 3. The existing global stations are far more than those shown in Figure 3, especially in the azimuth of about 100°, and an average can become very large. I do not know exactly what an averaging scheme is used. In this particular case, if we take a bin- average, we get the following:**

| Azimuth range | number of stations | bin average |
|---|---|---|
| 0-45 | 20 | 7.536 |
| 45-90. | 4 | 8.468 |
| 90-135 | 14 | 8.539 |
| 135-180 | 8 | 8.273 |

**The average of the bin average is 7.85, very close to the ISC value of 7.82. This is a new problem with recent events with so many stations, and the averaging scheme is very important.**

[Figure]

20021103  Denali  MS_ISC=7.82, $M_{S\_GCMT}$=8.6

[Figure]

Fig. 3

*Ditto.*

This bring us to a problem with old events again. As discussed on line 238 and illustrated in Table A1 of this manuscript, the famous 1906 San Francisco is given MS_ISC=8.61. However, as shown in Figure 4, only 6 stations are used, with 4 stations essentially in the same azimuth, and it is hard to assess the uncertainty of the assigned MS value. Since the Gutenberg notepad lists station MS values from some 16 stations, and Gutenberg and Richter (1954) gave MS= 8 1⁄4. I like this notation which probably implies an uncertainty of 1⁄4 magnitude unit. Unfortunately, many journals demand that it should be written as 8.25 which has a very different implication for the uncertainty. Also, I suspect that Gutenberg and Richter's assignment of "quality, A, B, and C, or a, b, and c is somewhat subjective on the basis of their experience, but in case of this kind of data to which rigorous statistical method is hard to apply, I believe that it is a very reasonable practice. Actually, Lienkaemper (1984) examined Gutenberg's notepad data, and came up with M=8.3. Although I did not follow exactly what he did, he did use the 16 stations. I suspect that the data listed in the Gutenberg's note pad did not meet the strict criterion of ISC, and were not adopted in "MS_Dataset" (I may be wrong.). However, as shown in the examples presented in Figures 1 to 4 in this review, the azimuthal station coverage is so important for obtaining a reasonable average that I would like to see as many station data as possible in the ISC catalog, even if the measurement procedure was slightly different. Overall, my experience is that the difference caused by the limited azimuthal coverage is far greater than that caused by the difference in the station practice.

[Figure]

190604  San Francisco  $M_{S\_ISC}=8.61$, $M_{\_GR}=8\,¼$, $M_{LK}=8.3$

Fig. 4

*We provide the MS uncertainty for every earthquake in the dataset. We have Abe's adaptation of Gutenberg's notepads but, unfortunately, we cannot find the Gutenberg's solution for the 1906 San Francisco earthquake. In Lienkaemper (1984) the list of stations contributing to MS (first earthquake in Table 3 of https://doi.org/10.1785/bssa0740062357) includes also amplitudes from publications and not only from the original station bulletins, Hence, many of those stations are not included in our MS solution for the 1906 San Francisco earthquake. As far as we know, station bulletins are not available for many of those stations, and we believe that Gutenberg and the other quoted authors sourced those amplitudes in different ways. This is the reason for our solution to list less station magnitudes in this case. The result, however, does not change much as most station magnitudes are well above 8, and this would still lead to a large discrepancy between MS and Mw from the literature. Hence, azimuthal effects alone cannot explain the large MS of this earthquake. We indeed suggest that the large MS may be due to instrumental issues.*

**My comments above are in no way the criticism of the ISC practice and catalog, but they represent my hope that the ISC catalogs will be used most effectively and properly by serious users. The historical data are important but they can have all kinds of problems and uncertainties, and very often rigorous handling is not possible. Nevertheless, these data do contain historical information which we cannot get otherwise. After all, how to use the data base and interpret it is ultimately the responsibility of the users, rather than the catalog producers, but it is most important that the catalog producers provide adequate caveats to the users so that the catalogs can be carefully used for understanding the Earth's seismicity.**

*We agree with this final remark.*

[revised manuscript text omitted]

---

## Author Comment (AC4)

**Author's Response to Emmanuel Scordilis (RC3) (PAPER: https://doi.org/10.5194/essd-2021-266)**

Domenico Di Giacomo & Dmitry A. Storchak

November 2021

We thank the reviewer #3, Emmanuel Scordilis for his comments and suggestions. Below we reply in detail to each point showing the Referee Comments (RC) in bold and the Author Respon se (AR) in italic. The revised version of the manuscript (annotated manuscript after our answers) is appended as well.

**RC:**

**A. General Comment**

**Earthquake catalogs, extending over a wide period and covering the globe are useful tools for many studies. Two are the critical preconditions that must be fulfilled: accuracy in their focal parameters and homogeneity regarding the scale in which their magnitudes are expressed. Considering the fact that it is not suffering saturation but only at its large values, Ms is a suitable magnitude for such studies.**

**In this spirit, I believe that this work is very useful and it is my sense that its outcome (the catalog) is going to be extensively used in the future.**

**The paper is well written and its content corresponds to its title. There are some minor issues that I will describe below, which, if clarified, I believe will further improve the manuscript.**

**Concluding, it is my opinion that the manuscript can be accepted for publication after some minor revision.**

*We thank the reviewer for the positive general comments.*

**Following are my comments in details.**

**B. Specific Comments**

1. **In the 1st paragraph of "Introduction" the basic pros of surface wave magnitude, Ms, are mentioned. I believe that the cons (e.g. inability of Ms estimation by using records of short period instruments and, therefore, of small local earthquakes, possible underestimation for very strong earthquakes) should be mentioned too.**

   *We have added the following sentences in the Introduction:*

   *"However, as any magnitude type, MS has also shortcomings, such as the possible underestimation for some large earthquake (as discussed later), the*

*inability of processing surface waves from short-period instruments (hence for many small local earthquakes) and the limitation, at least in standard procedures (IASPEI, 2013), of being defined for shallow earthquakes."*

2. **Page 3: In the square root, the factor "2" must be out of the brackets:**

$$\left(\frac{A}{T}\right)_H = \sqrt{2 \cdot \left(\frac{A}{T}\right)^2_{N/E}}$$

*We thank the reviewer for pointing out the typeset error.*

3. **How have you estimated the final surface wave magnitude if more than one Ms values were available? Mean value? Weighted mean? Have you applied any filters to avoid contamination that could be caused by one or more potentially incorrect magnitude values that may deviate significantly from the majority of the rest?**

*We do not average MS computations from different sources, but recompute MS using the station data available to us. This is outlined in detail in the text.*

4. **The final catalog includes only events with recomputed Ms magnitudes, meaning that this catalog is not complete, as it possibly misses earthquakes which could be included in catalogs published by other authors, covering wide regions and extending over wide time periods, but with magnitudes consistent (not original) to the standard Ms, (e.g. Karnik 1996).**

*This is true to some extent, and we acknowledge the fact the dataset presented here can be improved given time and resources. However, we have strong reasons to list only earthquakes that are backed up by station data (hence we say "recomputed MS"). We are aware of the work of Karnik and all earthquakes that 1) have station data to validate the occurrence of an earthquake, 2) allow relocation and MS recomputation, if enough station data is available, are included in our dataset. The requirement of station data is of paramount importance as, particularly in the pre-digital period, earthquakes from different sources contain errors, at times significant. A striking example is the case of the fake M8.2 Peru earthquake in 1908 studied by Di Giacomo and Dewey (2020). Hence, we avoid for this dataset to include earthquakes as listed from other sources where we do not have data or not enough quality data to reprocess the event and obtain our own instrumental solution.*

5. **In lines 72-74 you mention that:** ***"The locations adopted in this work come from the ISC‑ GEM Catalogue (Bondár et al., 2015; Di Giacomo et al., 2018) between 1904 and 1963 and the rebuilt ISC Bulletin (Storchak et al., 2017, 2020) from 1964 onward"*. Looking in the ISC-GEM catalog I could not find some earthquakes included in your catalog. Indicatively I mention the following events: 1904‑12‑02, 02:19:12; 1904‑12‑11,**

**17:05:42; 1908–01–31, 04:49:15 etc. These events are also not included in the online ISC bulletins. Figure 3 clearly shows that data before ~1950 are coming exclusively from ISC. So, which is their origin?**

*The ISC-GEM Catalogue is composed of two files, one for the main catalogue and a supplementary file listing earthquakes with low quality location and/or low quality Mw. Most of the earthquakes in the supplementary file have no Mw at all but it can happen that Mw, as converted from MS, is of low quality and hence an earthquake is not considered good enough to be listed in the main catalogue file. Such details are described in Di Giacomo et al. (ESSD 2018). The earthquakes mentioned by the reviewer are all in the supplementary file of the ISC-GEM Catalogue and not yet included in the ISC Bulletin. Therefore, we do not consider necessary to change the text.*

6. **Chapter 4 entitled "Catalogue Properties" gives a detailed and very useful analysis of the time-history of Ms scale. However, it looks there are four gaps regarding earthquakes of Ms<6.0 that can be observed in figure 10: one at ~1920, the second during 1940-1950, the third between 1960 and 1978 and the fourth between 1980 and 1984. The authors are right about the impact of World War 2, which justifies the second gap. According to the authors (see chapter 6), there is further work to be done that will possibly allow some of the above gaps to be covered. So, consider this as just a remark.**

*To answer also a remark by another reviewer we have extended the discussion on the fluctuations of the MS dataset content in different years and added the following sentences regarding the period 1960-1977:*

*"The period 1960-1977 also features less earthquakes below 5.5 than previous and following decades. This is due both to the limited number of stations available and the fact that we digitized surface wave data from the 1960s printed station bulletins only for earthquakes selected in the first version of the ISC-GEM Catalogue (magnitude 5.5 and above, Storchak et al., 2013). In Section 6 we propose activities that are likely to mitigate significantly the deficiencies of the ISC MS dataset in most of the 1960s-1970s."*

7. **In the same chapter (4) and in lines 160–170 there is an analysis of the features of the formed catalog. It is mentioned there how the completeness magnitude, Mc, is distributed over time. There is a point here that, in my opinion, needs clarification. To proceed to a meaningful Mc estimation and to study its variation with time it is necessary to know first if there are earthquakes systematically missing from the data set. I mean, are there any earthquakes whose focal parameters are known but they are not included in the catalog because it was not possible to have Ms estimation for them? If yes, then I believe that the term "completeness magnitude" should be avoided as, at least literally, it has another meaning.**

*All earthquake catalogues have missing earthquakes. However, Mc estimations are still useful to emphasize strengths and weaknesses of an earthquake catalogue. The purpose of our analysis is just that, and we feel that it has value for the reader and dataset user. Nevertheless, we are confident that we do not systematically miss earthquakes with magnitude above 6 from the 1920s-1930s and 7+ from 1905 (meaning some poorly recorded individual earthquakes may be missing, but not systematically). Also, apart from a few exceptions, we are confident that at this point if we were not able to recompute MS then the earthquake is likely below the Mc estimation provided at a given time. In addition, please note that we include earthquakes relocated with depth down to 60 km, and other catalogues may have different depths for the same earthquake which may lead to MS being allowed or not. Still, we think that an Mc estimation is possible and useful even if earthquakes are missing.*

8. **Figure 11 & Lines 180-190 (a follow up of the previous comment): The rates shown in figure 11 do not necessarily show variation of completeness magnitudes over time.**

*Our aim with Figure 11 is to update and compare the seismicity rate estimations from previous works with our dataset. We consider magnitude thresholds quite high (MS 6, 6.5 and 7), where we can be more confident that our dataset is mostly complete. As such, we feel that Figure 11 carries important information and address some misleading results in previous papers.*

9. **Figure 12: The Ms underestimation for very strong earthquakes (e.g. M>8.0) has been already observed and noticed (e.g. Heaton et al., 1986). For the example of the magnitude of the Aleutian earthquake of April 11, 1946, the magnitude reports in the ISC bulletin are: Ms=7.3 (after Abe, "Phys. Earth planet. Interiors", 1981); Ms=7.1 & Mw=8.0 (after Pacheco & Sykes, "Bull. Seism. Soc. Am.", 1992); Mw=8.6 (after López & Okal, "Geophys. J. Int.", 2006). These values are clearly showing (as the authors of this manuscript state) that Ms values underestimated the real magnitude of this great event. Therefore, although the data in figure 12 are not many, it can be stated that the "saturation" of Ms for values over ~8.0 is confirmed here and should therefore be considered as a fact.**

*In line with a remark by another reviewer we have slightly changed the discussion regarding the MS "underestimation" and point out that the largest differences occur for the so-called tsunami earthquakes, like the 1946 Aleutian earthquake. However, we reaffirm that this more an extreme occurrence rather the rule, hence we stand by our choice of suggesting of speaking of "MS underestimation" rather than "MS saturation".*

**C. Technical Corrections**

**I could not make it to locate in the text the following two references (included in the "References"):**

1) **Line 339: Bormann (2012)**

*This reference to Bormann (2012) is in Figure 2 caption.*

**2) Line 350: Di Giacomo and Storchak (2016)**

*This reference to Di Giacomo and Storchak (2016) is in Figure 1 caption.*

[revised manuscript text omitted]

---

## Referee Report (RR1)

**Report**

The authors have adequately answered most of my questions / comments in the revised version of the manuscript.

I still have reservations and objections about whether the Mc estimate is realistic when earthquakes are missing from the data set (lines 168-182 and 179-183 in the revised manuscript). In my view, Mc's estimate of a catalog, which contains all available data, is clearly more reliable than Mc's estimate of the same catalog, when some data, though existing, was omitted simply because does not meet certain criteria.

In closing, I think the manuscript can now be accepted for publication.

---

## Author Response (AR2)

**List of changes to RC3 (Emmanuel Scordilis) (PAPER: https://doi.org/10.5194/essd-2021-266)**

Domenico Di Giacomo & Dmitry A. Storchak

22 December 2021

Dear Editor,

we took into account the final comments by RC3 by adding in the revised manuscript the following sentences (pg 6 lines 170-175 in the revised manuscript):

"The completeness analysis is only meant to highlight general features of the dataset (a more detailed study in this respect, as the one by Michael, 2014, is not the aim of this work). As any other earthquake catalogue, we cannot include all shallow earthquakes since some of them may be poorly recorded or not matching our selection criteria. Hence, we perform annual Mc estimations knowing that the dataset may have missing records. This is more likely to be the case in the first part of the last century. As a result, annual Mc uncertainties (error bars in the top panel of Fig. 10) are generally larger for the early decades of the last century compared to recent times."

We believe we have addressed to final remarks, we hope this is satisfactory and the paper can be accepter for publication in ESSD

With Kind Regards

Domenico Di Giacomo and Dmitry A. Storchak